# Haplotype variability in mitochondrial rRNA predisposes to metabolic syndrome
Petr Pecina[1], Kristýna Čunátová [1], Vilma Kaplanová[1], Guillermo Puertas-Frias [1], Jan Šilhavý[2], Kateřina Tauchmannová[1], Marek Vrbacký[1], Tomáš Čajka [3], Ondřej Gahura [4], Markéta Hlaváčková[5], Viktor Stránecký[6], Stanislav Kmoch[6], Michal Pravenec[2], Josef Houštěk[1], Tomáš Mráček [1] ✉ & Alena Pecinová [1] ✉

Metabolic syndrome is a growing concern in developed societies and due to its polygenic nature, the genetic component is only slowly being elucidated. Common mitochondrial DNA sequence variants have been associated with symptoms of metabolic syndrome and may, therefore, be relevant players in the genetics of metabolic syndrome. We investigate the effect of mitochondrial sequence variation on the metabolic phenotype in conplastic rat strains with identical nuclear but unique mitochondrial genomes, challenged by high-fat diet. We find that the variation in mitochondrial rRNA sequence represents risk factor in the insulin resistance development, which is associated with diacylglycerols accumulation, induced by tissue-specific reduction of the oxidative capacity. These metabolic perturbations stem from the 12S rRNA sequence variation affecting mitochondrial ribosome assembly and translation. Our work demonstrates that physiological variation in mitochondrial rRNA might represent a relevant underlying factor in the progression of metabolic syndrome.

Mitochondria, the organelles of bacterial origin, still maintain mitochondrial DNA (mtDNA), and its sequence exerts variability (single nucleotide polymorphisms – SNPs), dividing the population into mtDNA families known as haplogroups[1]. Unlike nuclear DNA (nDNA), mtDNA in mammals is maternally inherited and encodes only 13 structural proteins, the subunits of the oxidative phosphorylation apparatus (OXPHOS), two ribosomal RNAs and 22 transfer RNAs required for mitochondrial protein synthesis. The nuclear genome encodes the remaining more than 1000 mitochondrial proteins, which are transported into the mitochondria[2]. This requires coordination in expression and translation between the two genomes[3,4]. Recently, studies in human populations have associated mtDNA haplogroups with the risk of metabolic syndrome or its complications[5–7], and sequence variation in mtDNA has also been shown to influence the expression of stress response genes[8].

A higher risk of metabolic syndrome is strongly associated with weight gain due to high nutrient excess[9]. The resulting aberrant distribution of fat, mainly in visceral and intra-abdominal depots[10,11], leads to lipotoxicity—ectopic fat storage in metabolically active tissues such as the liver, myocardium, or skeletal muscle[12]. The metabolic disbalance is associated with symptoms such as insulin resistance, type 2 diabetes mellitus (T2DM), non-alcoholic fatty liver disease, or cardiovascular disease[13].

While the link between mitochondrial function and metabolic syndrome is widely accepted, the molecular mechanisms behind it still need to be better defined. Many studies have associated insulin resistance and T2DM with a decrease in mitochondrial oxidative capacity[14,15], which results in the inefficient fatty acid oxidation. Subsequent accumulation of bioactive lipids such as diacylglycerols (DGs), fatty acyl-CoA, or ceramides have been shown to inhibit the insulin signalling pathway[16,17]. Specifically, it has been observed that an increase in plasma fatty acids results in elevated intracellular fatty acyl-CoA and DGs in skeletal muscle, leading to activation of protein kinase C θ (PKCθ). Subsequent phosphorylation of insulin receptor substrate 1 (IRS-1) prevents insulin-stimulated tyrosine phosphorylation and decreases insulin-stimulated glucose transport[17]. In contrast to DGs accumulation, high ceramide concentrations directly affect the activity of the serine/threonine protein kinase Akt/PKB, either through activation of phosphatase 2A[16,18] or through protein kinase C ζ (PKCζ)[19].

Another possible mechanism linking mitochondrial dysfunction to insulin resistance is the mitochondrial generation of reactive oxygen species (ROS). An increased supply of electrons from excess nutrients increases the

[1]Laboratory of Bioenergetics, Institute of Physiology, Czech Academy of Sciences, Prague, Czech Republic. [2]Laboratory of Genetics of Model Diseases, Institute of Physiology, Czech Academy of Sciences, Prague, Czech Republic. [3]Laboratory of Translational Metabolism, Institute of Physiology, Czech Academy of Sciences, Prague, Czech Republic. [4]Institute of Parasitology, Biology Centre, Czech Academy of Sciences, České Budějovice, Czech Republic. [5]Laboratory of Developmental Cardiology, Institute of Physiology, Czech Academy of Sciences, Prague, Czech Republic. [6]Research Unit for Rare Diseases, Department of Pediatrics and Adolescent Medicine, 1st Faculty of Medicine, Charles University, Prague, Czech Republic. ✉e-mail: tomas.mracek@fgu.cas.cz; alena.pecinova@fgu.cas.cz

likelihood of their slip to molecular oxygen, forming superoxide and, subsequently, other forms of ROS, which may directly induce insulin resistance, or damage mitochondrial proteins, lipids or DNA, thereby reducing oxidative capacity[20].

Insulin resistance has also been associated with low-grade inflammation in obese and diabetic patients[21–23]. This condition is characterised by chronically elevated inflammatory markers such as TNF-α, interleukin-1β or interleukin-6[24]. Also, mice overexpressing the chemokine ligand CCL2 (also known as MCP1) in adipose tissue exhibit insulin resistance[25]. Moreover, elevated cytokine levels correlated with increased lipolysis in adipose tissue, suggesting destabilised lipid droplets as a possible mechanism allowing better access of lipases to triacylglycerols[26].

Therefore, we asked whether the physiological variation in the mitochondrial DNA sequence may directly contribute to symptoms of metabolic syndrome. For this purpose, we used the unique model of conplastic rats carrying mtDNA from spontaneously hypertensive rat strain (SHR), Brown Norway strain (BN) or Fischer strain (F344) on the identical nuclear background of SHR strain that is widely used as an animal model of the metabolic syndrome[27–30]. SHR, BN and F344 strains harbour different mtDNA variants, which represent different mtDNA haplogroups present across the palette of laboratory rat strains[28]. The conplastic strains were derived by the multiple backcrossing of male SHR and female BN or F344 strain that allows to reach more than 99.8% identity in nuclear genome and thus isolate the effect of naturally occurring variation in the mitochondrial genome[27–29].

The previous studies demonstrated that the exposure of mtBN or mtF344 strain to high-fructose diet can promote systemic metabolic disturbances including glucose intolerance and/or elevated insulin levels during oral glucose tolerance test compared to control mtSHR strain[27,29]. In particular, the Brown Norway mtDNA variant led to a selective decrease of cytochrome *c* oxidase at protein as well as enzyme activity levels[29]. Furthermore, the analysis of metabolic phenotype of the left ventricles, muscle, and liver of mtF344 strain revealed reduced activity and content of several respiratory chain complexes. This associated with cardiac remodelling and changes in heart functional parameters compared to the mtSHR progenitor strain[27]. Nevertheless, the direct link between the disturbances of mitochondrial function and metabolic phenotype has not been described, yet. In the current study, we found that both conplastic strains developed insulin resistance on a high-fat diet, which in the mtF344 animals can be explained by differences in mitochondrial 12S rRNA sequence, leading to reduced substrate oxidation and subsequent accumulation of bioactive lipids and insulin resistance.

## Results

Our previous work has established and characterised rat conplastic strains, which differ only in their mtDNA[27,29,31]. Numerous SNP variants exist in mtDNA across the strains (summarised in Supplementary tables S1, S2). Overall, they affect 9 genes encoding for subunits of oxidative phosphorylation complexes, 7 tRNAs, and 12S as well as 16S rRNAs. As our preceding data indicated that conplastic strains differ in their metabolic phenotypes, we asked to what extent mtDNA sequence variability may affect insulin sensitivity and whether a high-fat diet could further aggravate such phenotype. Therefore, we designed an experiment to compare SHR rats (mtSHR) with conplastic strains possessing mtDNA from F344 or BN strain on the SHR nuclear background (mtF344 or mtBN, respectively). After weaning (week 5), the animals were transferred to either chow (CHD) or high-fat (HFD) diet for 15 weeks. The oral glucose tolerance test (OGTT) was performed at week 18 (2 weeks before experiment termination and tissue collection) to assess insulin sensitivity. The whole experimental design is summarised in Fig. 1a.

### The high-fat diet promotes insulin resistance in conplastic rats

As presumed, the HFD diet led to an increase in mean body weight in all the strains. Relative to mtSHR strain, the body weight of mtF344 was increased on either of the diets (Fig. 1b, S1a), and this was not caused by increased

caloric intake (Fig. 1c). As expected, the area under the curve (AUC) for glucose during OGTT was higher on HFD compared to CHD in all groups, indicating a lower ability to clear glucose from the blood (Fig. 1d, S1b). Also, insulin levels were increased significantly 30 min after glucose gavage on HFD (Fig. 1e). Interestingly, while AUC for glucose did not differ in mtF344 strain compared to mtSHR on either of the diets, insulin concentration was significantly higher in mtF344 animals on HFD. On the other hand, in the mtBN strain, glucose tolerance was decreased, while insulin levels remained unchanged (Fig. 1d, e).

We calculated homoeostatic model assessment (HOMA-IR) to quantify insulin sensitivity from fasting glucose and insulin levels (Fig. 1f). As expected, HFD led to a significant increase in HOMA-IR in all groups compared to CHD. Importantly, we also observed a significant genotype-dependent increase of HOMA-IR in mtF344 animals on HFD and in mtBN on both diets. These data indicate that relative to mtSHR, both mtF344 and mtBN mtDNA variants predispose to impaired insulin sensitivity on a high-fat diet.

### The high-fat diet-induced insulin resistance in mtF344 correlates with the accumulation of diacylglycerols

As a next step, we focused on the possible mechanism(s) responsible for differences in insulin sensitivity between conplastic animals. In principle, three different mechanisms were proposed to link mitochondrial function and obesity-induced insulin resistance: (i) elevated levels of inflammatory cytokines, (ii) increased oxidative stress, (iii) accumulation of bioactive lipids such as ceramides or DGs resulting from reduced oxidative capacity of mitochondria.

First, we focused on inflammatory response and assessed proinflammatory cytokine Il1b and chemokine Ccl2 levels in the liver and white adipose tissue. As expected, the expression of the genes was increased in animals on HFD (Fig. 2a, b, d, e), and the concentration of Il1b, but not Ccl2, measured by multiplex immunoassay was also increased on HFD in the liver (Fig. 2c, f). However, no difference between genotypes was observed in expression profiles. In summary, the changes in inflammatory response do not explain the strain-specific differences in insulin sensitivity.

Since both nuclear and mitochondrial genomes encode the subunits of mitochondrial oxidative phosphorylation complexes, their steric incompatibility may affect the stability of the complexes and, consequently, increase the production of reactive oxygen species (ROS). Therefore, we measured $H_2O_2$ production by Amplex red assay in isolated rat liver mitochondria. To distinguish where the electrons may escape towards oxygen, we used a combination of specific substrates and inhibitors of respiratory chain dehydrogenases (Fig. 3a). Using NADH-linked substrate (glutamate and malate) and rotenone, we determined the ROS generation on the flavin site of complex I ($I_F$, Fig. 3b). Compared to mtSHR, ROS production was unchanged in mtBN strain and was slightly lower in mtF344 on CHD.

Under the conditions of high flux from succinate oxidation and high proton-motive force, the electron backflow to complex I occurs. Under such conditions, electrons can escape to molecular oxygen at the Q site ($I_Q$), which may be prevented by rotenone[32,33]. Using saturating succinate concentration (10 mM) and complex I inhibitor rotenone (1 μM), we assessed rotenone-sensitive ROS production. We did not observe any changes in ROS generation between strains or diets (Fig. 3c).

Further, we measured ROS production on the flavin of complex II (Fig. 3d) using a physiological concentration (0.4 mM) of succinate. As expected, since complex II subunits are encoded solely by nDNA, we did not detect any changes in ROS production between strains. Similarly, no differences were detected with succinate and an inhibitor of complex III (antimycin A), which would reveal the changes in ROS production on complex III (Fig. 3e). These data indicate that mitochondria-derived oxidative stress cannot explain the strain-specific differences in insulin sensitivity.

Therefore, we focused on the content of bioactive lipids, such as ceramides or diacylglycerols, which have also been shown to modulate insulin

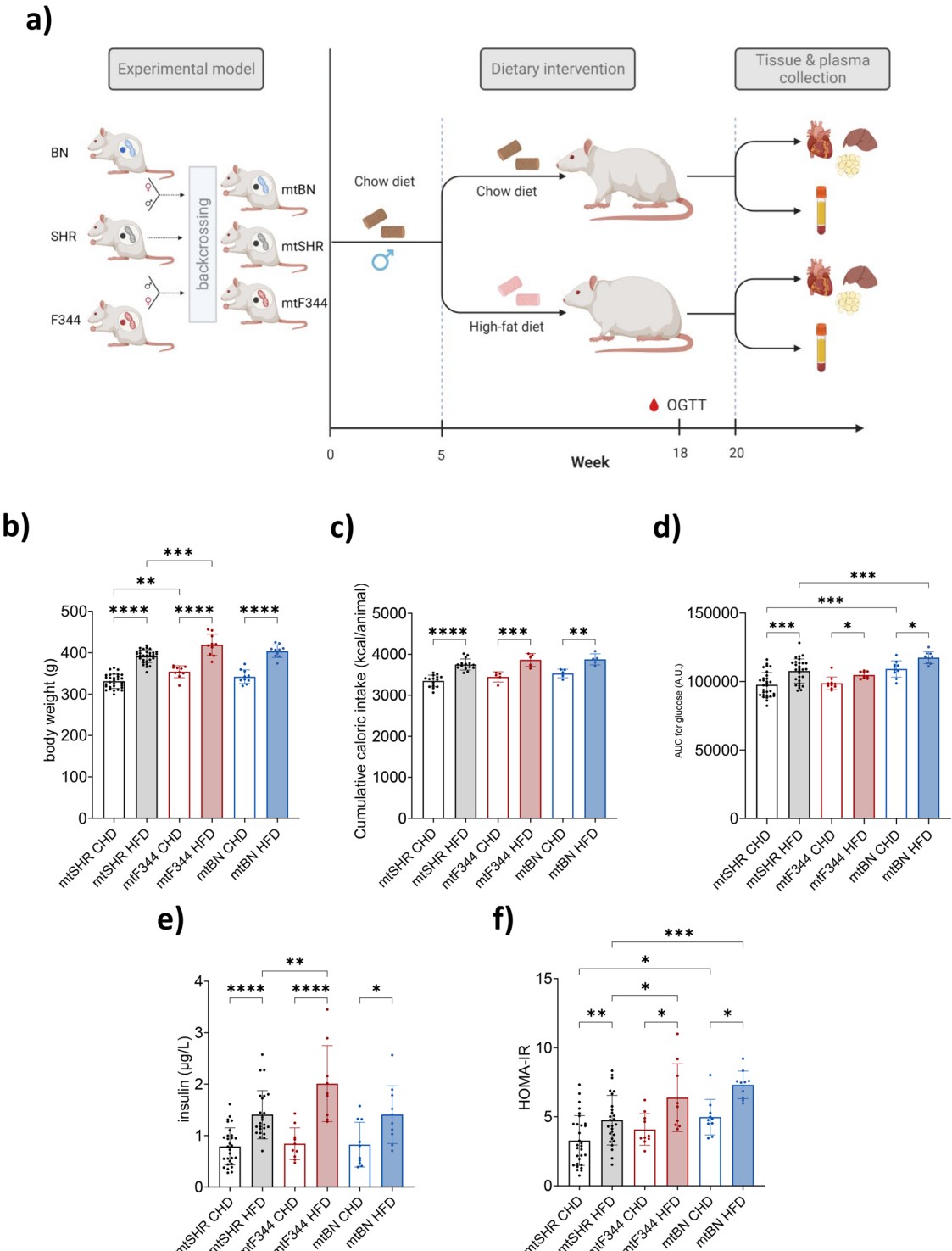

**Fig. 1 | Experimental design and metabolic phenotype of conplastic rats.**
**a** Schematic depiction of conplastic strains created by multiple backcrossing of males SHR rats with females of SHR, and females of progeny derived from (♀BN x ♂SHR) F1 or (♀F344 x ♂SHR)F1 hybrids. After weaning, experimental groups were transferred to a chow or high-fat diet for 15 weeks. At week 18, the OGTT test was performed, and at week 20, the tissues were collected for further analyses. Body weights (**b**) and cumulative caloric intake (**c**) after 15 weeks of dietary intervention. **d** Area under the curve calculated from oral glucose tolerance test. **e** Insulin levels 30 min after glucose gavage and (**f**) homoeostatic model assessment (HOMA-IR) calculated from fasting glucose and insulin levels. Data represent means ± SD from at least 9 animals. Asterisks represent $p$-value: * <0.05; ** <0.01; *** <0.001 **** <0.0001.

signalling. For this purpose, we performed unbiased lipidomic profiling of plasma, liver, and heart in all strains on HFD. Lipidomic profiles showed distinguished clusters on a sparse Partial Least Squares Discriminant Analysis (sPLSDA) for individual conplastic strains (Fig. 4a). We did not find increased levels of ceramides in conplastic strains relative to mtSHR.

The ceramides were even decreased—in all analysed matrices for the mtF344 (Fig. 4b) and in mtBN only in heart (Fig. 4c). However, in the mtBN strain, we observed a slight but significant increase of triacylglycerols (TGs) in the liver and heart and a slight increase of diacylglycerols (DGs, Fig. 4c). Interestingly, in mtF344 animals, DGs were elevated not only in the heart

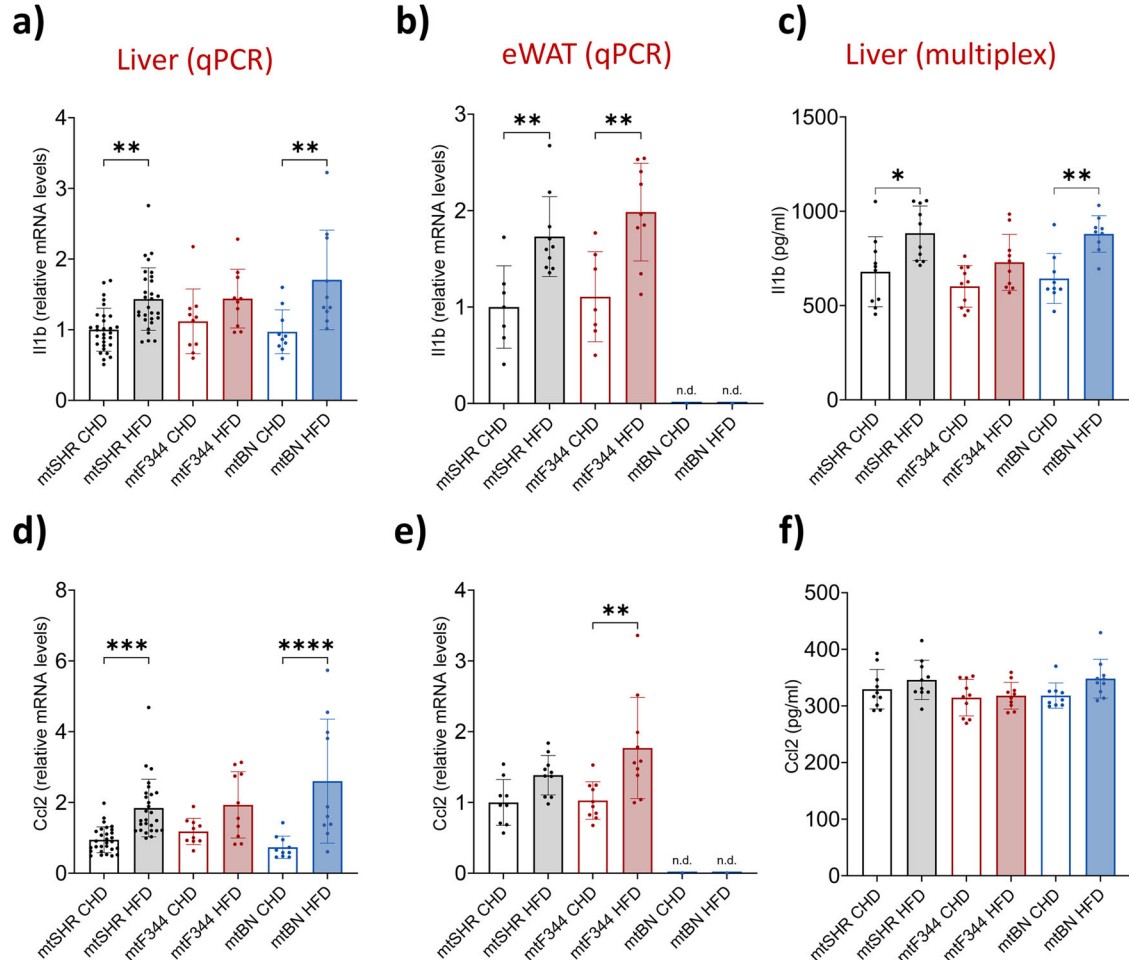

**Fig. 2 | Inflammatory response in conplastic rats.** Relative mRNA levels of pro-inflammatory cytokine *Il1b* (**a**, **b**) and chemokine *Ccl2* (**d**, **e**) in the liver (**a**, **d**) or epididymal adipose tissue (eWAT, **b**, **e**). Corresponding Il1b (**c**) and Ccl2 (**f**) levels in liver extracts measured by bead-based immunoassay. Data represent means ± SD from at least 7 animals. Asterisks represent *p*-value: * <0.05; ** <0.01; *** <0.001 **** <0.0001.

but also in the liver and plasma (Fig. 4b). The most abundant DGs, especially in plasma, contain acyl chains that are the most abundant in the diet (Fig. S2) – oleic acid (18:1), palmitic acid (16:0), and linoleic acid (18:2). Thus, it is likely that fatty acids stored in DGs originate from diet and did not undergo subsequent remodelling. The results suggest that DGs may serve as modulators of insulin signalling in mtF344 animals.

### mtF344 rats on HFD have lower mitochondrial oxidative capacity in the heart

We hypothesised that DGs accumulation may arise from defective mitochondrial fatty acid oxidation. Therefore, we focused on mitochondrial function and measured mitochondrial respiration in tissue homogenates (Fig. 5a). The capacity to oxidise fatty acids was assessed with palmitoyl-carnitine and malate as substrates – a significant HFD-induced increase in fatty acid oxidation capacity (FAO capacity) was observed in the liver of all strains (Fig. 5b). Total phosphorylating capacity (OXPHOS capacity) was then measured with a combination of complex I and II substrates, and also tended to be higher after HFD but became statistically significant only in the mtSHR group (Fig. 5c). Ultimately, the maximal electron transport chain (ETC) capacity was comparable between genotypes and diets (Fig. 5d). The data suggest that liver mitochondria possess a relatively high spare capacity of ETC independent of the mtDNA genotype and may utilise this capacity when fatty acid abundance increases. Therefore, a specific increase in fatty acid oxidation capacity should arise from the upregulation of enzymes involved in mitochondrial β oxidation. Indeed, LFQ proteomic analysis of liver tissue confirmed that many proteins involved in fatty acid oxidation are

increased, especially the enzymes required to oxidation of long chain fatty acids (Fig. S3a–c) while there is no profound change in the content of OXPHOS protein subunits (Fig. S3d–f).

In contrast to liver, such diet-dependent phenotype was not observed in the heart, where FAO capacity and overall OXPHOS capacity did not differ between diets in any of the groups (Fig. 5e–g). Accordingly, neither the levels of proteins involved in fatty acid metabolism were changed by HFD in mtSHR or mtF344 (Fig. S4). On the other hand, in heart we observed genotype-specific differences in oxidative capacity. Specifically, in mtBN on CHD, OXPHOS capacity was significantly increased. More strikingly, there was a consistent drop in all evaluated parameters (i.e. FAO and overall OXPHOS capacity as well as total ETC capacity) in mtF344 animals on HFD (Fig. 5e–g).

These data imply that HFD represents a stress factor for the mtF344 strain, leading to decreased substrate oxidation in both coupled (FAO/OXPHOS capacity) and uncoupled (ETC capacity) states. Also, in comparison to liver mitochondria, which are metabolically flexible, the heart mitochondria cannot further increase fatty acid oxidation capacity. This implies that mtF344 strain utilises a most of ETC capacity for ADP phosphorylation already on CHD. To compare reserve capacity of the respiratory chain (available for the higher flux demand) between tissues, we assessed the relative spare capacity (RS$_P$), which represents a ratio between spare capacity (i.e. ETC – OXPHOS) relative to OXPHOS capacity. RS$_P$ capacity in the liver was even higher than OXPHOS capacity (approximately 120% of OXPHOS capacity). It decreased after HFD (significantly only in the mtBN), which is in agreement with increased flux from fatty acid

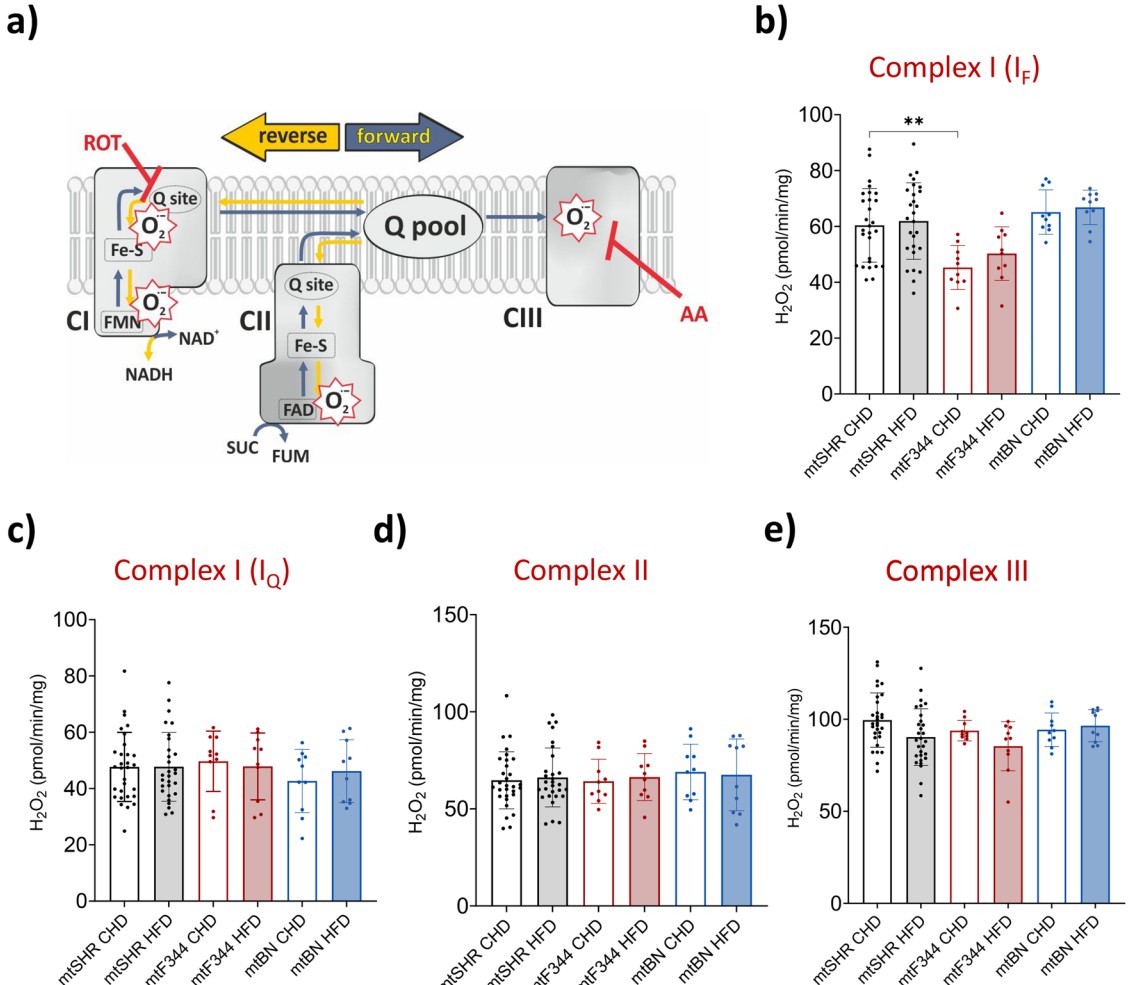

**Fig. 3 | ROS generation in conplastic rats. a** Scheme of superoxide production in electron transport chain (CI – complex I; CII – complex II; CIII – complex III; ROT – rotenone, complex I inhibitor; AA – antimycin A, complex III inhibitor; $O_2^{\cdot-}$ - superoxide). **b–e** Production of $H_2O_2$ by isolated liver mitochondria using Amplex Red assay, and specific substrates and inhibitors to distinguish ROS production site. Isolated mitochondria were supplemented with 10 mM glutamate, 2 mM malate and 1 μM rotenone (**b**), 10 mM succinate and 1 μM rotenone (**c**), 0.4 mM succinate (**d**) or 10 mM succinate plus 1 μg/mL antimycin A (**e**). Data represent means ± S.D. from at least 10 animals. Asterisks represent *p*-value: * <0.05; ** <0.01; *** <0.001 **** <0.0001.

oxidation and the subsequent higher requirement to utilise the ETC capacity (Fig. 5h). $RS_P$ capacity was lower in mtF344 strain on both diets (approximately 80% of OXPHOS capacity), yet still sufficient to yield an increase in oxidative capacity on HFD in liver.

On the other hand, the $RS_P$ in the heart was much lower than in the liver (approximately 40% of OXPHOS capacity) and was practically independent of diet (Fig. 5i). This suggests that in the heart response to changes in substrate availability cannot stem from functional adjustments in the respiratory chain. The unchanged $RS_P$ in the mtF344 group thus reflects that HFD led to a decrease in both OXPHOS and ETC capacities. Since we observed significant genotype-associated changes in respiratory parameters only in the mtF344 animals, in the subsequent experiments we only focused on mtF344 and characterisation of molecular mechanisms underlying this phenotype.

**Lower oxidative capacity in the heart of mtF344 rats on HFD is caused by decreased content of OXPHOS complexes**

In the next step, we focused on the factors responsible for decreased respiratory capacity on HFD. First, we performed LFQ proteomics and analysed proteome-wide responses in the liver and heart, focusing on the OXPHOS apparatus. In the liver, we observed a slight yet significant increase in the overall content of mitochondrial proteins on CHD and no difference on HFD (Fig. 6a). Detailed analysis of subunits of OXPHOS complexes

revealed a slight rise in some of the OXPHOS subunits in the mtF344 group on CHD but a decrease of complex I and III subunits on HFD (Fig. 6c). Decreased abundance of OXPHOS subunits was more pronounced in the heart on HFD, where not only subunits of all OXPHOS complexes were reduced (Fig. 6b, c), but even the content of all mito-proteins dropped (Fig. 6b). These data indicate that the lower respiratory capacity observed in the heart of mtF344 animals on HFD is caused by reduction of the OXPHOS proteins.

Supposedly, mitochondrial content is decreased on HFD in the heart of mtF344 animals relative to mtSHR controls. To confirm this hypothesis, we measured mtDNA copy number and citrate synthase activity, established markers of the mitochondrial content in both tissues. As depicted in Fig. S5, mtDNA copy number was comparable between genotypes as well as diets in the liver, and citrate synthase was slightly decreased in mtF344 on CHD (Fig. S5a). On the other hand, both parameters were suppressed in mtF344 heart tissue (Fig. S5b), supporting the results from LFQ analysis.

Subsequently, we checked whether the reduced mitochondrial content resulted from accelerated mitochondrial autophagy. We assessed the autophagosome marker LC3B normalised to citrate synthase in the heart, but we found that the mitochondrial autophagy was significantly decreased in mtF334 tissue on both diets (Fig. S6a), thus excluding autophagy involvement.

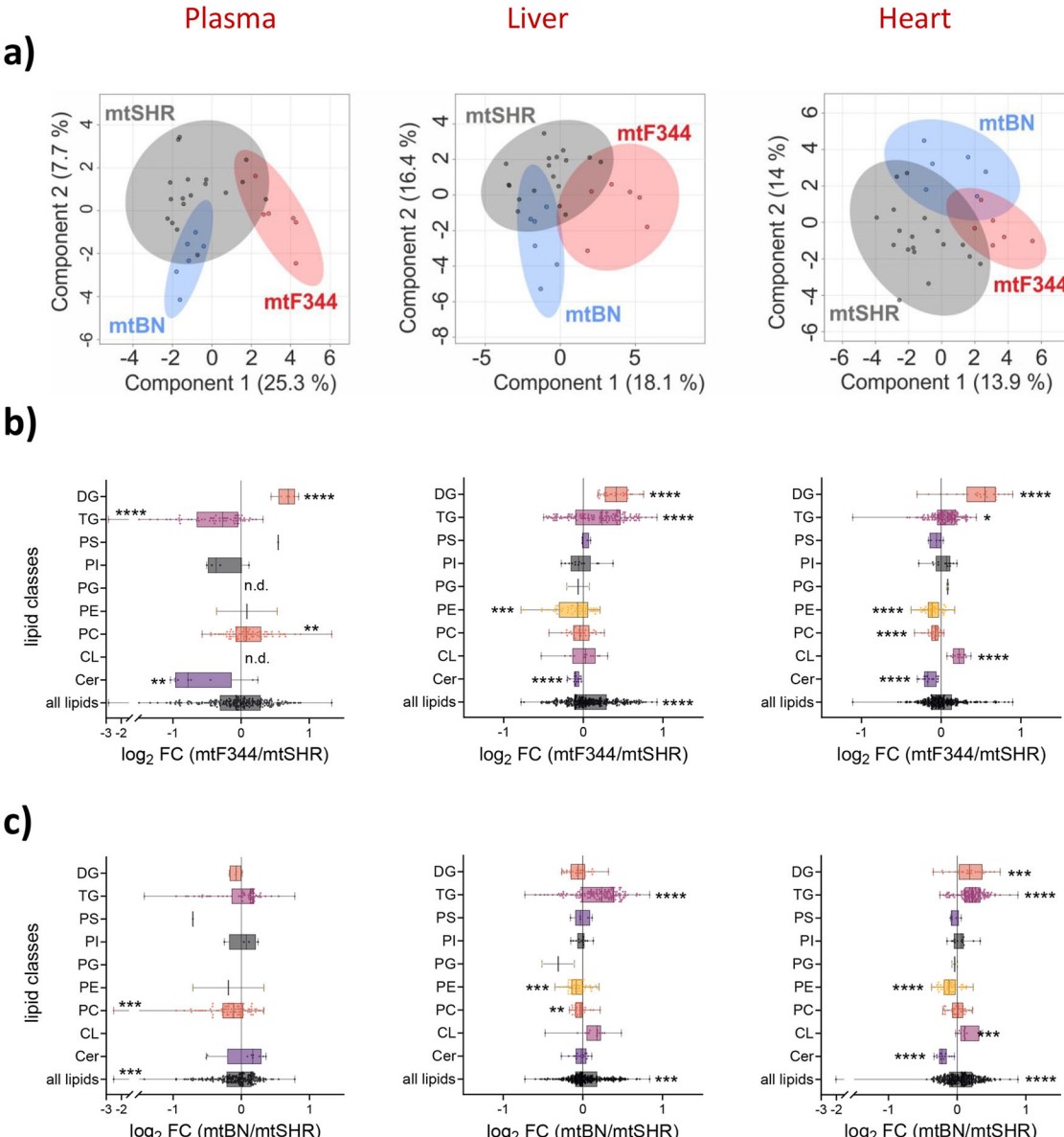

**Fig. 4 | Analysis of lipid classes in conplastic strains on HFD. a** sPLSDA was performed to differentiate between mtF344, mtBN and mtSHR strains. Each data point represents one animal. **b, c** LC-MS analysis of different lipid classes in plasma, liver, and heart of mtF344 (**b**) and mtBN (**c**) strain compared to mtSHR control group. The data are expressed as boxplots (median is depicted) of log₂ fold change from 6 animals. (Lipid classes annotation: Cer, ceramides; CL: cardiolipins; PC:

phosphatidylcholines; PE: phosphatidylethanolamines; PG: phosphatidylglycerols; PI: phosphatidylinositols; PS: phosphatidylserines; TG: triacylglycerols; DG: dia-cylglycerols). The significances were calculated as one sample t-test compared to the mtSHR control group ($n = 6$), and asterisks represent $p$-value: * <0.05; ** <0.01; *** <0.001 **** <0.0001.

To explore potential mechanisms behind the OXPHOS decrease in the heart on HFD, we performed a GO enrichment analysis of proteins significantly changed in the LFQ dataset. It identified pathways related to mitochondrial translation as most significantly enriched (Fig. 6d). Since the higher levels of N-formylmethionine (fMet) were shown to decrease mitochondrial protein synthesis in a haplotype-dependent fashion in humans[34], we assessed fMet levels by untargeted LC-MS approach. We identified significantly higher fMet levels on HFD relative to CHD in mtSHR animals (Fig. S6b). However, under the same conditions, the levels of OXPHOS subunits were not significantly decreased (Fig. S6c). Furthermore, fMet levels were markedly reduced in mtF344 compared to the mtSHR on HFD (Fig. S6b), which again did not correlate with the levels of OXPHOS proteins, which were decreased under the same conditions (Fig. 6c). Based on these data, we conclude that fMet levels do not regulate mitochondrial proteostasis in our model system.

Then, we focused on the mitochondrial translation apparatus, and checked the levels of the mitochondrial ribosome proteins by LFQ. We found pronounced decrease in proteins of the small mitochondrial ribosomal subunit (mtSSU) in the mtF344 group, especially on HFD (Fig. 6e). We also observed an increased content of proteins from the large mitochondrial ribosomal subunit (mtLSU) that was diet-independent in both tissues (Fig. 6e). Strikingly, we observed the same pattern of downregulated mtSSU and upregulated mtLSU subunits also when comparing hearts from progenitor SHR and F344 strains (Fig. 6e). Since F344 progenitors and mtF344 only share mtDNA, it strongly implicates that polymorphisms in mtDNA and not nuclear background are responsible for the observed downregulation in the content of mtSSU subunits.

To rule out the possibility, that the changed levels of mtSSU/LSU as well as OXPHOS proteins do not stem from the decreased transcript levels for individual proteins, we quantified transcripts of selected proteins from

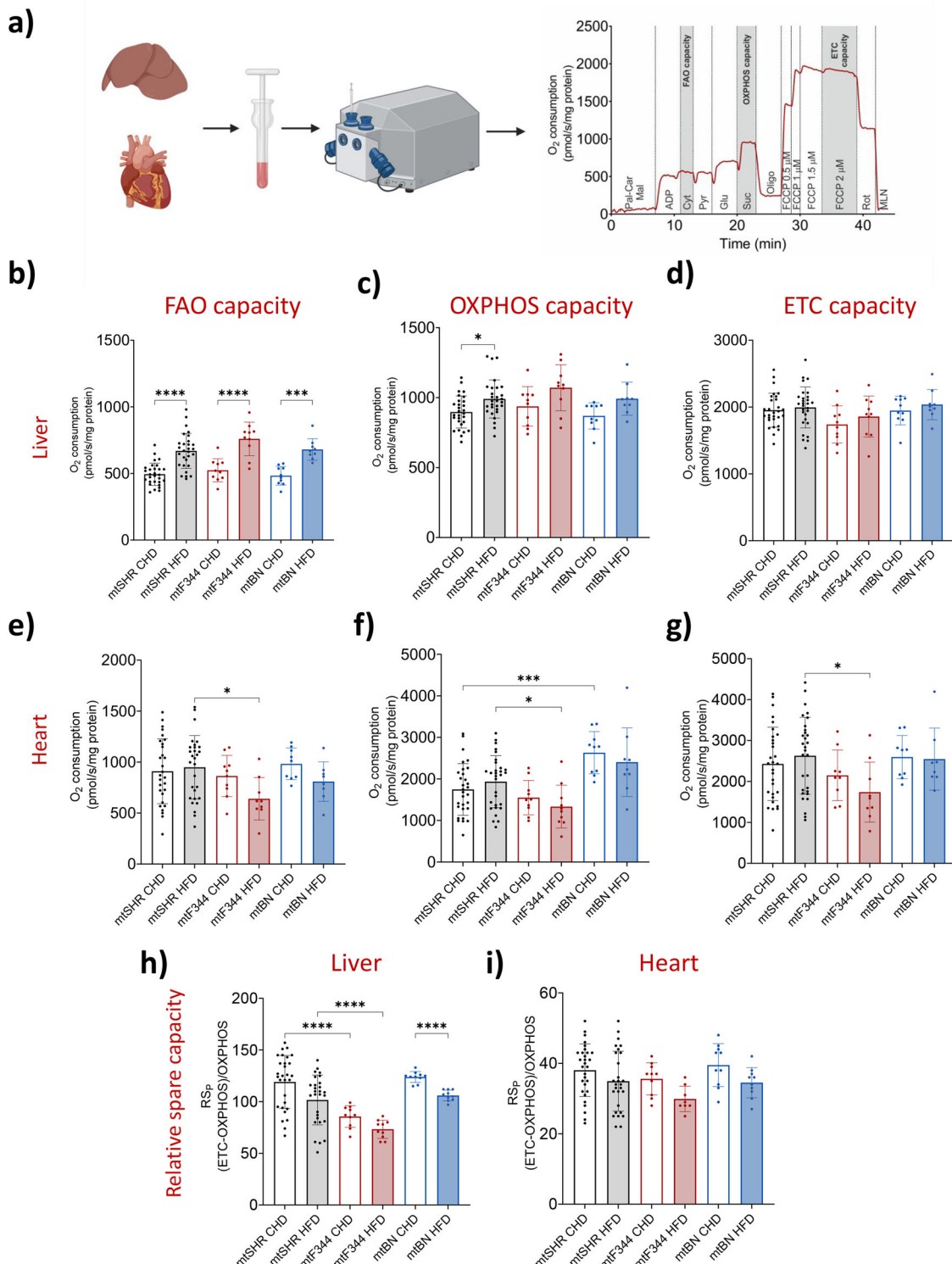

**Fig. 5 | Mitochondrial respiration in conplastic strains. a** Experimental workflow of the mitochondrial oxygen consumption in the liver and heart homogenates measured by Oxygraph-2k (Oroboros). The representative curve depicts the consecutive additions of different substrates and inhibitors (for more details, see Methods). The liver (**b, c**) and the heart (**e, f**) fatty acid oxidation (FAO) and OXPHOS capacities were measured in the presence of 1 mM ADP and either 50 μM palmitoyl carnitine and 2 mM malate (FAO capacity, **b** and **e**) or in the combination with 10 mM pyruvate and glutamate (OXPHOS capacity, **c, f**). ETC capacity (**d, g**) was determined by combining all substrates in the presence of uncoupler FCCP (1 μM). Relative spare capacity of ETC to OXPHOS capacity (RS$_P$) in the liver (**h**) and heart (**i**) was calculated as percentage of OXPHOS capacity, i.e. 100*(ETC – OXPHOS capacity)/ OXPHOS capacity. The data are expressed as means ± SD from at least 8 animals. Asterisks represent p-value: * <0.05; ** <0.01; *** <0.001 **** <0.0001.

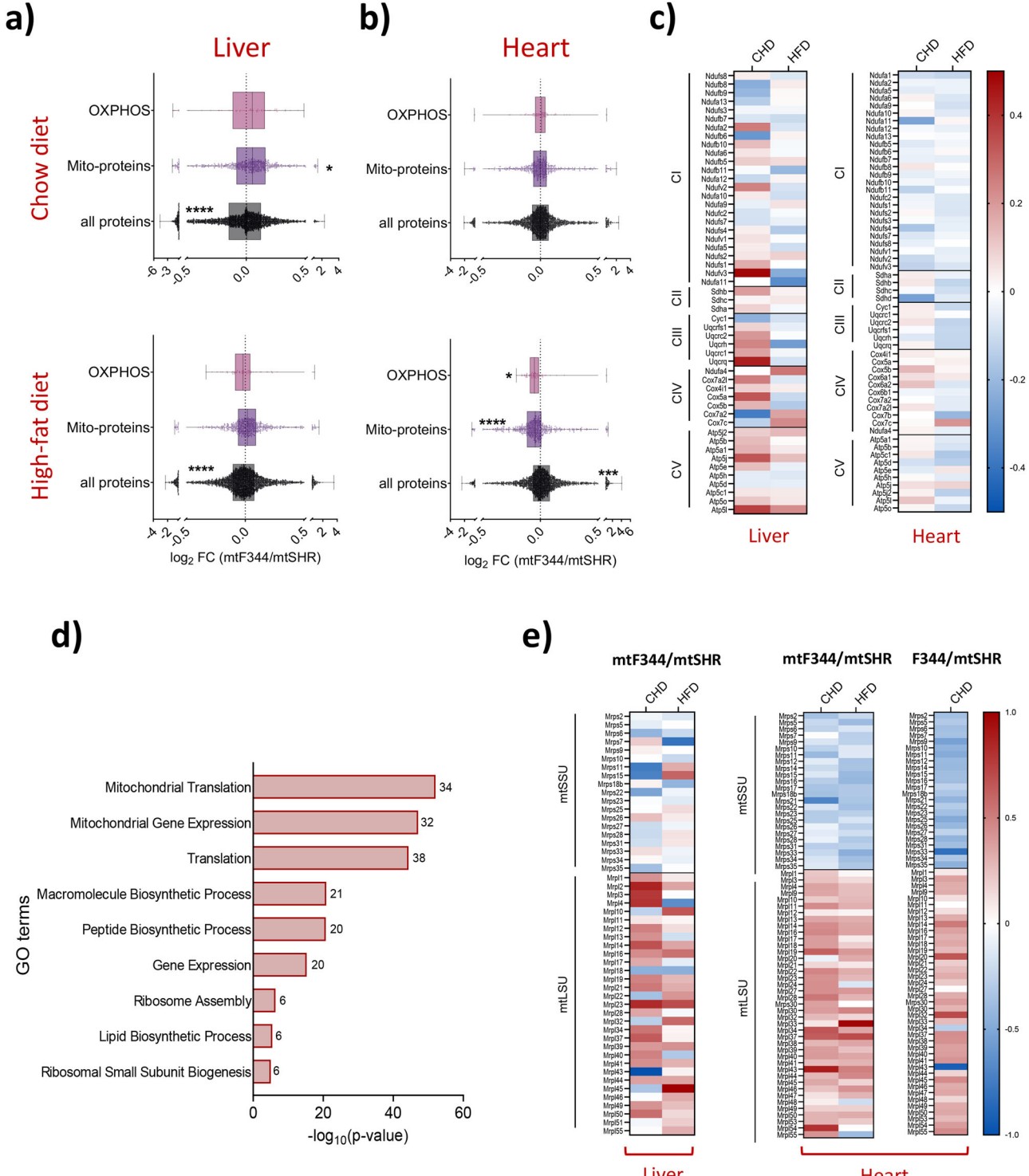

**Fig. 6 | Analysis of protein levels in the liver and heart in mtF344 strain.** LFQ-MS analysis of liver (**a**) and heart (**b**) of mtF344 compared to mtSHR control group. Data are expressed as boxplots (median is depicted) of all proteins, mitochondrial (Mito-proteins – MitoCarta3.0 annotated proteins) and OXPHOS (subunits of complex I–V). **c** Heat maps depicting the average $\log_2$ fold-change of individual subunits of OXPHOS complexes between mtF344 and mtSHR groups in heart on HFD. **d** GO enrichment analysis using differentially expressed proteins in the heart of mtF344 compared to the mtSHR group on HFD. Numbers represent gene counts in particular GO term. **e** Heat maps depicting the average $\log_2$ fold-change of individual proteins of mitochondrial small (mtSSU) or large (mtLSU) ribosome subunits between mtF344 compared to mtSHR (liver and heart tissue) and progenitor F344 and mtSHR (heart tissue). The significances in (**a**) and (**b**) were calculated as one sample t-test compared to the mtSHR control group ($n = 6$). Asterisks represent p-value: * <0.05; ** <0.01; *** <0.001 **** <0.0001.

mtSSU (Mrps22 and Mrps34), mtLSU (Mrpl37) and three mtDNA encoded OXPHOS subunits (mt-Nd1, mt-Cyb and mt-Co1). As depicted in Fig. S7, we did not observe any drop in the transcript content for mtDNA-encoded proteins. In the case of representative mitochondrial ribosomal subunits, we even observed significant increase in the *Mrps22* transcript, possibly representing some sort of compensatory mechanism, since at the protein level, Mrps22 was the most profoundly downregulated one (Fig. S7a). The data implicate that the OXPHOS proteins and mitochondrial content are

generally suppressed due to the reduced amount of mtSSU that is not caused by lower expression of the mtSSU proteins, but rather the mitochondrial ribosomal subunits biogenesis is affected.

### Downregulation of small mitochondrial ribosomal subunit proteins leads to the attenuation of mitochondrial proteosynthesis

Based on these data, we hypothesised that the drop in mtSSU proteins quantity may lead to slower synthesis of mtDNA encoded proteins. Thus, we performed in vivo metabolic labelling in rat skin fibroblasts derived from mtF344 and mtSHR animals to assess the rate of mitochondrial translation. After 3-hour incubation with DMEM medium containing $^{35}$S-methionine and $^{35}$S-cysteine, the amount of newly translated mtDNA encoded proteins was significantly decreased in mtF344 animals (Fig. 7a). This decrease affected all evaluated OXPHOS subunits, indicating a general drop in mitochondrial proteosynthesis expected for the system with reduced content of the mtSSU. Similarly to the heart tissue, the transcript levels of the mitochondrial ribosome genes were comparable between mtF344 and mtSHR animals. The transcript levels of three mtDNA encoded subunits were not changed (Fig. 7b) and the levels of all OXPHOS complexes with mtDNA-encoded subunits were slightly yet not significantly decreased in mtF344 fibroblasts (Fig. 7c). This suggests that the mitochondrial protein translation (not transcription) is slower but can be still sufficient in skin fibroblasts under non-stress conditions to achieve mitochondrial proteostasis. These results agree with our findings in the animals on a chow diet (i.e. under non-stress conditions), where we did not observe any changes in OXPHOS protein content (Fig. 6a–c).

Mammalian mtSSU is composed of 30 proteins and a 12S rRNA, encoded in the mitochondrial genome by the *mt-Rnr1* gene. Interestingly, compared to SHR strain, *mt-Rnr1* of F344 strain possesses two single nucleotide polymorphisms (Table S2). They are localised in the 3′ minor domain helix h44, which contains two adenines that are universally conserved in all ribosomes and play a key role in decoding of mRNA[35,36]. We compared the sequence of helix h44 of *mt-Rnr1* from the SHR and F344 strains and 7 other species (Fig. 7d). While the 5′ and the 3′ ends of h44, which form the proximal part of the stem involved in decoding, are highly conserved, the two polymorphisms are in the variable region corresponding to the distal part of the helix. Superposition of human, mouse and porcine mtSSU structures showed that the variable region (depicted in red in Fig. 7e) adopts slightly different structures in the three species, and it is relatively loose and exposed to the mtSSU surface on the intersubunit side. During the biogenesis of mtSSU, the variable part of h44 assumes its mature conformation only after dissociation of the assembly factor NOA1, which interferes with its position[37]. The polymorphisms might affect the process of h44 folding or play a role in an earlier, so far uncharacterized, phase of the assembly[35,37]. Together, these findings may indicate a possible interaction with yet unknown regulator, either during the translational cycle or during the mtSSU assembly.

We propose that the mtDNA sequence variance in *mt-Rnr1* of the F344 strain may affect mtSSU assembly or a step in the translational cycle, leading to slower mitochondrial protein synthesis and possibly represents a risk factor for insulin resistance and metabolic syndrome development.

In summary, single nucleotide polymorphisms in F344 mitochondrial ribosomal RNA led to the lower content of the mtSSU subunit of the mitochondrial ribosome. This limits the rate of mitochondrial proteosynthesis and, as observed in heart, under stress conditions (such as a high-fat diet), may cause a drop in the content of OXPHOS complexes. As a result, the overall OXPHOS capacity is compromised, causing inefficient oxidation of substrates including fatty acids, which subsequently leads to the accumulation of bioactive diacylglycerols. Accumulated diacylglycerols may then interfere with insulin signalling and ultimately manifest as an impairment in insulin sensitivity. These disturbances occur in a stress- and tissue-dependent manner, implying the role of OXPHOS protein translational thresholds across different tissues[38–40].

## Discussion

Metabolic syndrome represents a significant burden in developed societies. While its initial trigger in the form of increased body weight may not be a problem, it is associated with other symptoms, including insulin resistance, type 2 diabetes, cardiovascular diseases or cancer[41]. Notably, common variants of mitochondrial DNA (mtDNA) have been indicated as potential risk factors for symptoms of metabolic syndrome in the human population (summarised in ref. 42). However, the mechanism underpinning this link is still largely unknown. Furthermore, there may be numerous pathways linking primary mtDNA sequence variation to metabolic syndrome. Since mtDNA encodes the structural subunits of OXPHOS complexes, primary protein level variation will always stem from them. However, physiological response to variants in complex I may be different to variation at the level of complex III or IV. Furthermore, variability in tRNA or rRNA may affect mitochondrial translation in general and have further reaching consequences.

Due to the variation in nuclear DNA, the impact of naturally occurring single nucleotide polymorphisms (SNPs) in mtDNA in humans may be concealed. To isolate the effect of mtDNA difference, the models of conplastic cell lines[34,43,44], conplastic mice[45] and conplastic rats[27,29,31] possessing identical nuclear genome, but different mtDNA sequence have been developed. Compared to the majority of mouse strains that share the same mtDNA haplotype[28,46], the inbred rat strains are divided into four major mtDNA haplogroups[28,29]—Brown Norway (BN), Fischer F344 (F344), Lewis (LEW) and spontaneously hypertensive rat (SHR) strains. To analyse the contribution of mtDNA variants on complex pathophysiological traits in SHR, conplastic strains possessing the four different mtDNA haplotypes on the SHR nuclear background (mtSHR, mtBN, mtF344 and mtLEW) were derived. The strains differ only in single nucleotide polymorphisms in mitochondrial OXPHOS structural genes, tRNAs and rRNAs on the uniform nuclear background[27,29,31], summarised in ref. 30. The mtDNA variation in these strains was associated with a selective reduction in the content and activity of OXPHOS complexes and risk factors for type 2 diabetes mellitus (T2DM) when the animals were exposed to a high-fructose diet for two weeks[27,29–31].

Besides sugar-based diets, obesity-inducing diets highly enriched in fats (high-fat diet, HFD) mimic the dietary habits of the human population and therefore are widely used in animal experimental studies[47]. However, the literature focused on the role of naturally occurring mtDNA variations during HFD-induced obesity is limited. Comparison of mice with C57BL/6 nuclear genome and either progenitor mtDNA or mtDNA from NZB/OlaHsd strain revealed the changes in body weight and adipocyte size after three months on HFD[45]. Also, distinct mitochondrial genetic background significantly impacted the animal metabolic efficiency in the model of mitochondrial-nuclear exchange (MNX) mouse exposed to HFD for 6 weeks, in which the mtDNA from the C3H/HeN mouse was inserted onto the C57/BL6 nuclear background and vice versa[48]. Further, in the mouse strains that differ in mtDNA encoded subunits of complex I (*mt-Nd2* and *mt-Nd5*), the HFD induced changes in gut microbiota[49].

To provide comprehensive knowledge about the impact of common polymorphisms in mtDNA on metabolic phenotype in obesity, we exposed SHR rats possessing parental (mtSHR), F344 (mtF344) or BN (mtBN) mitochondrial genome to HFD for 15 weeks. In agreement with the data in fructose-fed animals[27,29], we found decreased glucose tolerance in the mtBN strain and increased insulin level 30 min after glucose administration in mtF344 (Fig. 1d, e). Furthermore, increased homoeostatic model assessment (HOMA-IR) implies that both conplastic strains are predisposed to impaired insulin sensitivity during obesity (Fig. 1f). The mechanisms proposed to explain HFD-induced insulin resistance include increased production of pro-inflammatory cytokines, higher oxidative stress or accumulation of bioactive lipids[50,51]. In our model, we identified that mtF344 animals accumulated diacylglycerols (DGs) in plasma, liver and heart (Fig. 4b). Increased DGs have been shown to increase phosphorylation of insulin receptor in a protein kinase C dependent manner, which ultimately leads to the development of insulin resistance[17]. In the mtBN strain, none of

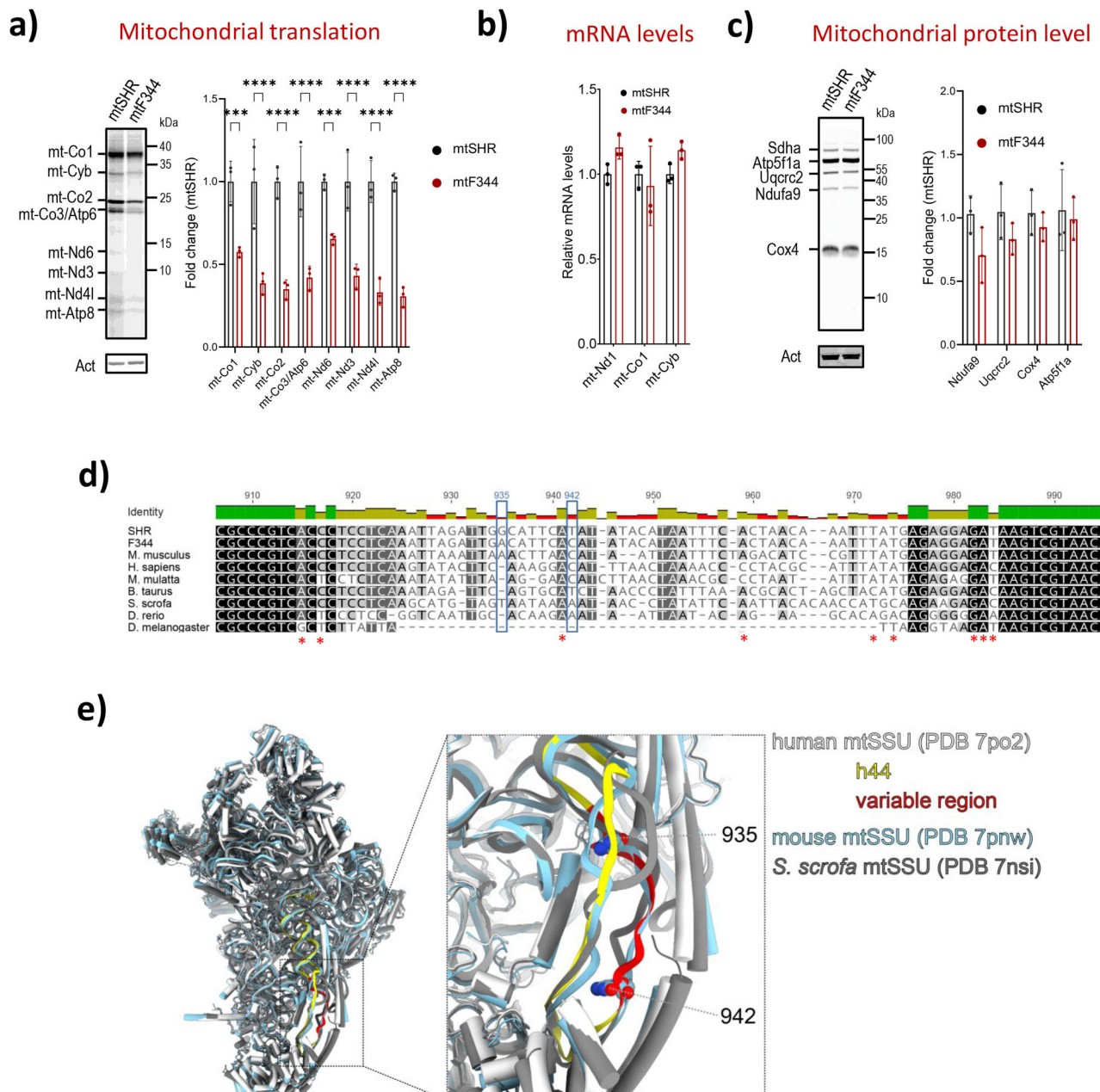

**Fig. 7 | Mitochondrial protein translation and the small mitoribosomal subunit.**
**a** Metabolic in vivo labelling with [35]S of mtDNA-encoded OXPHOS subunits after 3 h with [35]S-methionine and [35]S -cysteine. Representative image shows autoradiographic detection of labelled proteins in 60 µg protein of whole cell lysates analysed by SDS-PAGE. mt-Nd1, 3, 4 l and 6 – complex I; mt-Cyb – complex III; mt-Co1–3 – complex IV; mt-Atp6, 8 – complex V. Actin (Act) antibody was used as a loading control. **b** Relative mRNA level of OXPHOS subunits *mt-Nd1*, *mt-Co1* and *mt-Cyb*. **c** Representative Western blot analysis of the steady-state levels of subunits Ndufa9 (complex I); Uqcrc2 (complex III); Cox4 (complex IV), and Atp5f1a (complex V) normalized to Sdha (complex II). Actin IgM (Act) antibody was used as a loading control. Data represent means ± SD from 3 independent experiments. Asterisks represent p-value: * <0.05; ** <0.01; *** <0.001 **** <0.0001 (*n* = 3). **d** Multiple sequence alignment of h44 from selected species was generated by Clustal Omega[102] and visualised in Geneious Prime (Biomatters Ltd.). Sites of single nucleotide polymorphisms in F344 mtDNA are highlighted in blue rectangles, reported positions of human pathogenic variants indicated by the red asterisks. **e** Structures of human (white, PDB 7po2)[35], mouse (pale blue, PDB 7pnw)[35] and porcine (grey, PDB 7nsi)[103] small mitoribosomal subunits (intersubunit side) were superposed and visualised in ChimeraX[104]. The helix h44 of human mtSSU rRNA is shown in yellow, and the variable region is red, the position of SNPs in 12S rRNA of F344 strain (m.935 and m.942) is depicted.

the studied mechanisms was altered. However, liver and heart triacylglycerols were increased, indicating dyslipidemia in the tissues (Fig. 4c).

The accumulation of lipid intermediates could be caused by inefficient oxidation of metabolic substrates. Indeed, the studies in obese individuals suffering from insulin resistance demonstrated lower oxidative capacity[52] compared to healthy controls. It was also shown that overall respiratory chain activity measured as oxidoreductase activity of $NADH:O_2$ is reduced

in individuals suffering from T2DM[53]. In the current study, we observed tissue-specific differences in oxidative capacities. In agreement with published data[54,55], the fatty acid-dependent oxidative capacity in the liver of all three groups was significantly increased in HFD-fed animals when palmitoyl carnitine was used as a substrate (FAO capacity) and rather unaffected with the mixture of the substrate (compare Fig. 5b, c). Since the level of OXPHOS complexes is not changed or decreased after HFD, the data

indicate that flux of electrons from fatty acid β oxidation towards the respiratory chain is accelerated. Indeed, the levels of almost all proteins involved in β oxidation in the liver during HFD are elevated (Fig. S3).

While the increase of oxidative capacity of fatty acids during HFD is apparent in the liver, the published data addressing this issue in the heart are ambiguous. Using isolated heart mitochondria and palmitoyl carnitine as a substrate, the oxygen consumption was not changed in Wistar rats fed with HFD for three weeks[56] but decreased in C57BL6 mice on HFD for 24 weeks[57]. In the case of our models, diet did not affect neither the level of β oxidation proteins (Fig. S4), nor the mitochondrial respiration in the heart. However, the oxidative capacities were reduced in mtF344 animals compared to control animals fed with HFD (Fig. 5e–g). It indicates deterioration of the heart mitochondrial function caused by lipids overload that leads to DGs accumulation due to inefficient substrate oxidation[51] in the mtF344 strain caused solely by mtDNA sequence variation.

Analysis of the respiratory reserve capacity revealed that in the liver, the $RS_P$ (the capacity of the respiratory chain that is not utilised during ADP phosphorylation) is comparable to the OXPHOS capacity (Fig. 5h). This represents a reserve in the mitochondrial respiratory chain, which may get utilised during a higher substrate supply (e.g. from fatty acids). The $RS_P$ was significantly decreased in the mtF344 strain, but apparently it was still sufficient to deal with electrons originating from a higher rate of β oxidation. The $RS_P$ was much lower in the heart and did not reflect genotype or dietary intervention (Fig. 5i). As the heart uses fatty acids as the primary substrate to produce ATP[58], the data imply that heart mitochondria operate close to the maximal fatty acid-dependent oxidation, thus even a slight reduction of mitochondrial function may translate into observable phenotype. The different response to HFD in various tissues is also supported by the finding that the biochemical thresholds of OXPHOS complexes differ among the tissues[40]. For instance, the complex IV threshold that reflects excess enzyme activity is highest in the liver[40].

In the mtF344 strain, we demonstrated that mitochondrial mass, including OXPHOS proteins, is diminished in the heart of HFD-fed animals (Figs. 6b, c and S5b), which was not a consequence of increased autophagy (Fig. S6a) or suppressed gene expression (Fig. S7b). GO enrichment analysis of differentially expressed proteins identified mitochondrial translation as the most significantly changed process in the heart of mtF344 fed with HFD (Fig. 6d). Recently, changes in circulating N-formylmethionine (fMet) levels were associated with human mtDNA haplogroups. It was demonstrated that higher levels of fMet modulate mitochondrial and cytosolic protein synthesis[34]. This is of interest, since fMet initiates mitochondrial translation and may be a relevant player in our mtF344 animals. In response to HFD, the levels of fMet were increased in the heart of mtSHR. However, no change in the levels of the subunits of OXPHOS complexes was observed (Fig. S6b, c). Moreover, compared to mtSHR, in HFD-fed mtF344 animals, the decreased fMet levels and lower levels of OXPHOS proteins (Figs. 6, S6b) do not agree with published data[34,59,60]. Therefore, we concluded that fMet-mediated regulation of protein translation is not the cause of altered mitochondrial translation in mtF344 animals.

The key observation to dissect pathogenic mechanism was proteomic analysis that demonstrated reduced levels of the proteins constituting a small mitochondrial ribosomal subunit (mtSSU) in conplastic mtF344 animals in comparison to control mtSHR. Moreover, the comparison of progenitor F344 strain with control mtSHR showed analogous decrease of mtSSU proteins (Fig. 6e). Concomitantly, the mitochondrial protein synthesis was attenuated in skin fibroblasts derived from mtF344 animals compared to mtSHR (Fig. 7a), but the expression of the mitochondrial protein coding genes and mitochondrial ribosomal genes remained unchanged (Figs. 7b, S7c). Hence, the only genetic variability in components forming mtSSU remains the rRNA constituent. Between mtSHR and mtF344 strains, the *mt-Rnr1* gene that encodes for 12S rRNA harbours two mismatches – m.935 G > A (genome numbering, corresponds to gene number 868 G > A) and m.942 T > C (875 T > C) (Table S2).

The two polymorphisms are located within the helix 44 of the 12S rRNA 3′ minor domain (h44, Fig. 7d). In humans, h44 maturates during the formation of decoding centre, the key functional region of the mtSSU, and its folding is regulated by GTPase NOA1 which hinders the interaction of h44 with the docking site. Dissociation of NOA1 allows h44 to adopt its mature conformation[37]. The part of h44 at the base of its stem contributes to mRNA decoding during translation[61]. However, the two polymorphisms are located into a distal variable region of h44 that forms two parallel stretches of single stranded RNA exposed towards the mtSSU surface on the inter-subunit side (Fig. 7d, e)[62] and does not interact with any protein in mature mtSSU[63] or during translation[61]. Hypothetically, the region could interact with an unknown regulator. In the conplastic mtF344 model the mitochondrial translation is slowed down due to the reduced protein (not transcript) level of mtSSU proteins implying possible role of h44 variable region in the biogenesis of mtSSU.

Both *MT-RNR1* and *MT-RNR2* represent the most constrained sequences within human mtDNA with highest percentage of invariable bases[64]. Therefore, also pathogenic nucleotide substitutions in human *MT-RNR1* are relatively rare, and mostly associate with deafness. Only two (m.1555 A > G and m.1494 C > T) have been confirmed and several others have reported status in MITOMAP database (Fig. 7d, red asterisks) and they are the major contributors to aminoglycoside-induced and non-syndromic genetic hearing loss in patients with maternally inherited DEAFness, autism spectrum intellectual disability, possibly antiatherosclerotic[65]. Transmitochondrial m.1555 A > G cell lines also pointed to a role for mitochondrial protein synthesis[66], but in the absence of experimental models of mtDNA manipulation, the molecular mechanisms involved remained largely unknown[67–69]. These variants localize to the same h44 helix of *MT-RNR1* as the polymorphisms identified in mtF344 rats, yet predominantly DEAFness polymorphisms involve evolutionary more conserved nucleotides (Fig. 7d, e). It is therefore feasible, that mutations at more conserved positions have more severe DEAFness associated phenotype, while those at positions with lower degree of conservation may contribute to metabolic syndrome.

Interestingly, there is documented m.1518 C > T (871 C > T) polymorphism in human population mapping exactly to the same position as m.942 T > C polymorphism in mtF344 rats, which occurs at frequency of $10^{-4}$ to $10^{-5}$[65]. However, its potential link to metabolic syndrome would have to be established. Further, the analysis of mitochondrial rRNA variants in humans with deafness identified the mutation at position m.1517 A > C (870 A > C) that is neighbour to the m.1518 C position. As the m.1517 A is unpaired and not involved in any hydrogen bond, the variant was considered as silent mutation in the context of a mature mtSSU. The authors also performed structural analysis of the rRNA variants in the proximity of mito-ribosomal bridges, which revealed that m.1537 C > T (890 C > T) transition could affect the functioning of bridge mB5 in h44[62]. Interestingly, the m.1512 A (corresponds to m.935 A in rats) residue is very close to m.1537 C residue in tertiary structure, and the nucleotide transition may thus also influence the bridge functioning during translation. The potential link between *MT-RNR1* sequence variation and diabetic traits was suggested by the mitochondrial genome-wide association mapping of metabolomic phenotypes in humans. By this approach two prominent polymorphisms in *MT-RNR1* were identified (m.715 G > A and m.856 A > G), which associated with C2/C10:1 or SM (OH)C16:1/lysoPC a C28:1 ratio, respectively[70]. All these metabolites have links to regulation of insulin secretion and may therefore be analogous to the phenotype we identified in mtF344 rats. Ultimately, it should also be mentioned, that within *MT-RNR1* sequence 16AA microprotein MOTS-c is encoded, which is exported from mitochondria and appears to affect the regulation of cellular metabolism and insulin action in age-related diseases, such as type 2 diabetes mellitus[71]. However, this mechanism acts independently of mitochondrial protein synthesis.

In recent years, the research on the role of mtDNA variants in metabolic regulation and disease development accelerates[36,72]. Aw et al.[73] studied the effect of diet in 4 strains of *D. melanogaster* with different mitochondrial genomes and standardized nuclear genome (mitotypes). They observed switch in the relative fitness in two mitotypes on high protein or high

carbohydrate diet that was driven by the larval development and found that the changes are driven by naturally occurring point mutation in mtDNA encoded complex I subunit[73]. The crosstalk between mitochondrial and nuclear genomes in response to dietary intervention in *D. melanogaster* with diverse mitonucleogenotypes was also studied by another group[74]. They found that the SNP in the gene encoding for 16S rRNA (*mt:lrRNA*, m.13934 C > T) is sufficient to elicit the fitness impacts of altering diets which suggest that the variation in mtDNA does not need to change protein sequence to interact with nuclear genome[74]. In humans, the mutation in *MT-RNR2* encoding for 16S rRNA (m.1728 G > A) was also associated with the increase of tumorigenic potential of Hodgkin lymphoma cells[75].

Further, the mouse model of mitoribosomal mistranslation created by homozygous knock-in of mutant *Mrps5* that increases the error frequency rate of mitochondrial protein synthesis was studied[76–78]. The analysis of brain mitochondria revealed reduced basal and maximal respiration, suppressed levels of ATP and increased ROS generation[76]. The mutation led to hearing loss and age and stress-related alterations in behavioural traits in affected animals[76]. Moreover, the metabolic profiling of skeletal muscle demonstrated elevated levels of age-associated metabolites accompanied by increased glycolysis, pentose phosphate pathway and fatty acid synthesis indicating the alterations of specific bioenergetic processes in an age-dependent manner[77,78]. Another mouse models with mutated mitochondrial ribosomal protein Mrps12 leading to error-prone (K72I) or hyper-accurate (K71T) mitochondrial translation has been recently investigated[79,80]. The hyper-accurate translation resulted in slightly lower levels of overall mitochondrial proteins that did not have defects and consequently mitochondrial stress response was not activated to rescue protein synthesis. Consequently, the OXPHOS function was compromised and resulted in cardiomyopathy. When the animals were exposed to high-fat diet, enhanced translational accuracy protected the liver from lipid accumulation, but the animals still developed hypertrophic cardiomyopathy and the lifespan was reduced. These results suggest that translational rate is more important than translational accuracy[79,80], which is in agreement with our findings where decreased rates of mitochondrial translation are associated with decreased oxidative capacity, specifically affecting the heart.

In summary, the rat conplastic models enabled unique detailed characterisation of physiological variants of the mitochondrial rRNA. The common variation in rRNA sequence leads to a lower rate of mitochondrial translation via the reduction of the mtSSU subunit of the mitochondrial ribosome. Under the stress conditions such as high-fat diet, it results in lower expression of mtDNA encoded proteins and inefficient oxidation of substrates, including lipids. Consequently, accumulated bioactive diacylglycerols may lead to the impairment of insulin signalling. For the first time, we show that the common variants in mitochondrial *mt-Rnr1* may represent a risk factor for insulin resistance.

## Methods

### Animals

Animal experiments were approved by the Institutional Animal Care and Use Committee and the Committee for Animal Protection of the Czech Academy of Sciences (Approval Number: 58/2021) in agreement with the Animal Protection Law of the Czech Republic, which is fully compatible with the guidelines of the European Community Council directives 2010/63/EU. We have complied with all relevant ethical regulations for animal use. All efforts were made to minimise animal suffering and to reduce the number of animals used. All animals were housed in controlled conditions (22 ± 2 °C, 12:12 h of light-dark cycle) with access to water and respective diets. The males of different strains were used in the study including progenitor highly inbred spontaneously hypertensive rats (SHR) (SHR/OlaIpcv strain) and highly inbred F344 (F344/Crl) strain as well as conplastic strains derived by selective replacement of the mitochondrial genome of SHR strain with the mitochondrial genome of highly inbred strains of Fischer (F344) or Brown Norway (BN). The conplastic strains were generated by sequential backcrossing of F344 (53 backcrosses) or BN (59 backcrosses) females with SHR males.

The high number of backcrossing will ensure highly identical nuclear genomes in conplastic strains, since by 10th backcross generation nuclear genome of conplastic and recipient strain is approximately 99.8%[28,30]. The ratio of heterozygous versus homozygous loci corresponds theoretically to $1.1 \times 10^{-16}$ (mtF344) and $1.73 \times 10^{-18}$ (mtBN). Whole Genome Sequencing analysis of heart samples from individual conplastic strains was performed to verify that the strains are homoplasmic for respective mtDNA variants and do not carry contaminating heteroplasmy from SHR progenitors (Fig. S8, Table S3). Thus, the three different strains harbour the SHR (mtSHR), F344 (mtF344) or Brown Norway (mtBN) mitochondrial genome on identical SHR nuclear genetic background[27,29] (Fig. 1a). Also, the animals of progenitor F344 strain was used.

### Metabolic phenotype

Each strain was split into two groups of ten individuals, and at the age of 5 weeks (1 week after weaning), we fed one group on a chow diet (CHD, 1310 from Altromin, 14% kcal as fat, energy density 3.2 kcal/g) and a second group by high-fat diet (HFD, D12451 from Research diets, 45% kcal as fat, energy density 4.7 kcal/g) for 15 weeks. Body weight and food intake were recorded every week and cumulative caloric intake (kcal/animal) was calculated from cumulative food intake (g) by one animal multiplied by the energy density of the respective diet.

At the end of the study, rats in the ad libitum fed state were anaesthetized with isoflurane (2%), blood and tissues were collected for final biochemical analyses, and the animals were sacrificed by cervical dislocation. Blood was centrifuged at 2000 *g* for 10 min (room temperature), and plasma was collected and stored at −80 °C.

### Oral glucose tolerance test and insulin measurement

An oral glucose tolerance test (OGTT) was performed at the end of the dietary intervention using a glucose load of 300 mg per 100 g of body weight after 16 h fasting. The glucose was administered by glucose gavage. Blood was drawn from the tail immediately before glucose administration (time point 0) and then after 30, 60, 120, 240, and 360 min timepoints, and the concentrations of glucose and insulin were measured. According to the manufacturer's instructions, glucose levels were determined using a Glucose (GO) assay kit (Merck) and the area under curve (AUC) was determined from these values in GraphPad Prism. Blood insulin concentrations were measured by Rat Insulin ELISA kit (Mercodia). The homoeostasis model for insulin resistance (HOMA-IR) was calculated according to[81] from fasting levels of insulin and glucose by the formula: HOMA IR = serum insulin (pmol/L) *blood glucose (mmol/L)/22.5.

### Gene expression analysis

The total RNA was isolated from the liver, heart, epididymal white adipose tissue or cultured rat skin fibroblasts using the RNeasy Plus Universal Mini Kit (Qiagen, Venlo, Netherlands), and cDNA was synthesised from 2 μg of RNA by reverse transcription (SCRIPT cDNA Synthesis Kit, Jena Bioscience GmBH, Jena, Germany). Gene expression assays for *Il1b* (Rn00580432_m1), *Ccl2* (Rn00580555_m1), *mt-Nd1* (Rn03296764_s1), *mt-Cyb* (Rn03296746_s1), *mt-Co1* (Rn03296721_s1), *Mrps22* (Rn01447174_m1), *Mrps34* (Rn01470434_gH), *Mrpl37* (Rn01417163_m1) and *Hprt1* (Rn01527840_m1) were carried out on a ViiA7 instrument (Thermo Fisher Scientific, Waltham, USA) with the following cycling protocols: 95 °C for 12 min and 40 cycles at 95 °C for 15 s, 60 °C for 20 s, and 72 °C for 20 s. All reactions were conducted in triplicate, and 1.5 μL of diluted (1:10) cDNA was used in each 5 μL reaction using HOT FIREPol Probe mix Universal (Solis Biodyne, Tartu, Estonia). Transcript quantity was calculated in Quant Studio SW (Thermo Fisher Scientific, Waltham, MA 02451, USA). *Hprt1* transcript levels were used as a housekeeper reference.

### Tissue homogenates and mitochondria isolation

Tissue homogenates (7.5–10%, w/v) were prepared at 4 °C in STE medium (0.25 M (for heart) or in 0.35 M (for liver) sucrose, 10 mM Tris-HCl, 2 mM

EDTA, pH 7.4 containing protease inhibitor cocktail, PIC 1:1000, P8340, Merck KGaA, Darmstadt, Germany) using glass-Teflon homogeniser and filtered through a fine mesh[82]. Heart samples were then re-homogenised with Dounce glass-glass homogeniser. All homogenates were used fresh for oxygen consumption measurements or were stored at −80 °C for other assays. For the hydrogen peroxide production assay, freshly liver mitochondria were isolated as described in ref. 82. Protein content was measured using the Bradford method[83].

### Rat skin fibroblast culture
Primary rat skin fibroblasts were prepared from 3 weeks old mtSHR, mtF344 and mtBN animals, according to[84]. Established cultures of skin fibroblasts were then maintained and subcultured at 37 °C and 5% $CO_2$ in air in DMEM medium (Thermo Fisher Scientific) that was supplemented with 10% foetal calf serum (Merck) and penicillin/streptomycin solution (Thermo Fisher Scientific).

### Inflammatory markers
The 10% tissue extracts were prepared from 50 mg of tissue (liver or epididymal white adipose tissue) by adding 450 µL of extraction buffer (PBS with 0.1% NP-40 and protease inhibitor cocktail 1:500). The extracts were milled in 2 mL tubes containing ceramic bead (30 Hz for 3 min) using the mill (MM400, Retsch) and centrifuged at 12000 $g$ for 10 min at 4 °C. The supernatants were used to assess inflammatory markers Il1b and Ccl2 using LEGENDplex Multiplex assay (BioLegend, San Diego, USA) according to the manufacturer's instructions.

### Hydrogen Peroxide Production
Hydrogen peroxide production was determined fluorometrically by measuring the oxidation of Amplex UltraRed (Thermo Fisher Scientific) as described in ref. 85. The assay was performed with 10 µg of liver mitochondria in a KCl-based medium (120 mM KCl, 3 mM HEPES, 5 mM $KH_2PO_4$, 3 mM $MgSO_4$, 1 mM EGTA, 3 mg/mL BSA, pH 7.2) supplemented with 0.1 mM phenylmethyl sulfonyl fluoride (PMSF) in order to inhibit carboxylesterase that unspecifically converts amplex red to resorufin[86]. To distinguish ROS production in different complexes of the electron transport chain, specific substrates and inhibitors were added. Complex I: 10 mM glutamate plus 2 mM malate and 1 µM rotenone (IF) or 10 mM succinate and 1 µM rotenone (IQ), complex II: 0.4 mM succinate or complex III: 10 mM succinate and 1 µg/mL antimycin A. Amplex UltraRed was used at the final concentration of 50 µM with horseradish peroxidase (HRP) at 1 U/mL. The fluorescence signal from the well containing all substrates and inhibitors, but not mitochondria, was subtracted as a background for every experimental condition used. The signal was calibrated using $H_2O_2$ at the final concentration of 0–5 µM, and $H_2O_2$ stock concentration was routinely checked by measuring its absorption at 240 nm.

### Metabolomics and lipidomics
Metabolomic and lipidomic profiling of plasma, liver, and heart samples was performed using a biphasic solvent system comprising of methanol, methyl *tert*-butyl ether, and 10% methanol for metabolite extraction[87,88]. This was followed by four liquid chromatography-mass spectrometry (LC-MS) platforms: (i) analysis of polar metabolites using hydrophilic interaction chromatography in positive ion mode; (ii) analysis of polar metabolites using reversed-phase chromatography (RPLC) in negative ion mode; (iii) analysis of complex lipids using RPLC in positive ion mode and (iv) analysis of complex lipids using RPLC in negative ion mode, with optimised conditions reported previously[89]. For data processing, MS-DIAL 4 software was used[90], including annotation of polar metabolites using an in-house spectral library combined with MS/MS libraries available from various sources (NIST20, MassBank.us, and MS-DIAL MS/MS library). In addition, complex lipids were annotated using in silico MS/MS spectra available in the MS-DIAL software. Data sets were exported for each matrix and platform as signal intensity from the detector (peak heights) and filtered by removing

metabolites with (i) a max sample peak height/blank peak height average <10, (ii) an $R^2 < 0.8$ from a dilution series of quality control (QC) sample, and (iii) a relative standard deviation (RSD) > 30% from QC samples injected between 10 actual study samples. The data were then normalised using locally estimated scatterplot smoothing (LOESS) with QC samples injected between 10 actual study samples. The data were analysed in Metaboanalyst 5.0[91].

### Oxygen consumption
The oxygen consumption was measured in liver and heart homogenates as described in ref. 92. Oxygen consumption was measured in the homogenate (0.05–0.15 mg/mL) at 30 °C using the Oxygraph-2k (Oroboros Instruments GmbH, Innsbruck, Austria). The respiratory substrates and inhibitors were used at the following concentrations: 50 µM palmitoyl carnitine, 2 mM malate, 10 mM pyruvate, 10 mM glutamate, 10 mM succinate, 5 µM cytochrome *c*, 1 mM ADP, 5–100 nM oligomycin, 1–3 µM FCCP, 1 µM rotenone, 10 mM malonate. The oxygen consumption rates were analysed using DatLab 5 software (Oroboros Instruments GmbH) and were expressed in pmol $O_2$/s/mg protein. Relative spare capacity of ETC to OXPHOS capacity ($RS_P$) was calculated as percentage of OXPHOS capacity, i.e. 100*(ETC − OXPHOS capacity)/ OXPHOS capacity.

### Proteomics
Liver and heart samples were pulverised in liquid nitrogen, solubilised in 1% SDS and processed according to the SP4 no glass bead protocol[93]. About 500 ng of tryptic peptides were separated on a 50 cm C18 column using a 2.5 h elution gradient and analysed in a DDA mode on Orbitrap Exploris 480 (Thermo Fisher Scientific) mass spectrometer equipped with a FAIMS unit. The resulting raw files were processed in MaxQuant v2.1.4.0. with the label-free quantification (LFQ) algorithm MaxLFQ using the rat proteome database (UP000002494_10116.fasta, UniProt Release 2022_01). Downstream analysis was performed in Perseus (v. 2.0.7.0). GO enrichment analysis of proteins significantly changed in the LFQ dataset was performed by Enrichr software (https://maayanlab.cloud/Enrichr/)[94]. Data were visualised in GraphPad Prism 10 as heatmaps or boxplots (the limits are 25th to 75th percentil, the line in the middle represents median, whiskers from min to max values). The mass spectrometry proteomics data have been deposited to the ProteomeXchange Consortium via the PRIDE[95] partner repository with the datasets' identifiers PXD045910 and PXD052976.

### mtDNA copy number
The total genomic DNA was isolated using the GeneAid Genomic DNA mini kit (Geneaid, New Taipei, Taiwan). The mitochondrial copy number was determined according to[96], using TaqMan assay for mitochondrial *mt-Nd1* gene and nuclear *Hbb-b2* gene (Rn03296764_s1, Rn04223896_s1, Thermo Fisher Scientific). Quantitative real-time PCR was carried out on a ViiA 7 instrument (Thermo Fisher Scientific, Waltham, MA 02451, USA) using HOT FIREPol Probe mix Universal (Solis Biodyne, Tartu, Estonia).

### Enzyme activities
Complex I and IV activities were determined spectrophotometrically in liver homogenates at 30 °C as cytochrome *c* oxidoreductases[82,97]. The assay medium for complex I contained 50 mM Tris–HCl, 1 mM EDTA, 2.5 mg/mL BSA, 1 mM KCN, pH 8.1 and 100 µM NADH. The reaction was started by adding 40 µM cytochrome *c* and changes of absorbance at 550 nm were monitored. The rotenone-insensitive (5 µM rotenone) portion was subtracted. Complex IV activity was measured in a medium containing 40 mM K-$P_i$, 1 mg/mL BSA, pH 7.0. The reaction was started with 30 µM reduced cytochrome *c*, and its oxidation was monitored at 550 nm for 40 s. Citrate synthase (CS) activity was determined in liver and heart homogenates using a medium containing 0.1 M Tris–HCl, 0.1 mM 5,5′-dithiobis-(2-nitrobenzoic acid), 50 µM acetyl coenzyme A, pH 8.1[82]. The reaction was started by adding 0.5 mM oxaloacetate and then monitoring changes at 412 nm for

1 min. The data were corrected for the absorbance change without oxaloacetate. Enzyme activities were expressed as nmol/min/mg protein using molar absorption coefficient $\varepsilon_{550} = 19.6$ mM/cm (complex I and IV) or $\varepsilon_{412} = 13.6$ mM/cm (CS).

## SDS-PAGE and Western blotting

Proteins separation under denaturing conditions was performed in heart homogenates or cultured fibroblasts using tricine-sodium dodecyl sulfate polyacrylamide gel electrophoresis (SDS-PAGE). The heart homogenates were mixed with the SLB buffer (sample lysis buffer; 2% (v/v) 2-mercaptoethanol, 4% (w/v) SDS, 50 mM Tris (pH 7.0), 10% (v/v) glycerol, 0.02% Coomassie Brilliant Blue R-250), and incubated at 65 °C for 10 min. The cells were washed with ice-cold PBS and harvested in ice-cold RIPA buffer (150 mM NaCl, 1% Nonidet NP-40, 1% sodium deoxycholate, 0.1% SDS, 50 mM Tris, pH 8.0) supplemented with Protease Inhibitor Cocktail (1:500, Merck P8340) and benzonase® nuclease (1:1000, Merck 70664). Samples were centrifuged (10,000 × g, 15 min), supernatant was mixed with the SLB buffer, and incubated at 65 °C for 10 min. 20 µg of tissue or cell samples were separated on 12% polyacrylamide gels using the Mini-PROTEAN III apparatus (Bio-Rad, USA) and transferred to a polyvinylidene difluoride (PVDF) membrane (Immobilon FL 0.45 µm, Merck) by semi-dry electroblotting (0.8 mA/cm$^2$, 1 h) using a Transblot SD apparatus (Bio-Rad)[98]. Immunodetection was performed using the OXPHOS kit (1:250, Abcam, ab110412) or primary antibodies against LC3B (1:500, Abcam, ab48394), citrate synthase (1:1000, Abcam, ab129095) and actin (1:6000, Calbiochem, CP01-1EA), and fluorescent secondary antibodies AlexaFluor 680 (1:3000, Life Technologies) and IRDye 800 (1:15000, LI-COR). Detection was performed using the fluorescence scanner Odyssey (LI-COR Biosciences) and the signals were analysed and quantified by Image Lab software (Bio-Rad). Experiments were performed three times to assess the statistical significance of the results.

## Metabolic labelling

The rate of mitochondrial protein synthesis in fibroblast cultures established from individual conplastic rat strains was investigated by metabolic labelling with $^{35}$S-methionine and $^{35}$S-cysteine in the presence of emetine, an inhibitor of translation on cytosolic ribosomes, essentially as described[99,100]. Cells grown to 80% confluency on a 60 mm culture dish were washed three times with PBS. 15 min incubation in DMEM medium without methionine and cysteine was followed by the addition of emetine (100 µg/mL). After 15 min, the medium was exchanged for DMEM medium supplemented with emetine and $^{35}$S-Protein Labelling Mix ($^{35}$S-Met and $^{35}$S-Cys, Perkin Elmer NEG072; 100 µCi/mL). Cells were incubated for 3 h at 37 °C, and 250 µM cold methionine and cysteine were added. After 15 min at 37 °C, cells were washed twice with PBS supplemented with 250 µM cold methionine and cysteine, once with PBS and lysed using 150 µL of RIPA solution (150 mM NaCl, 1% Nonidet NP-40, 1% sodium deoxycholate, 0.1% SDS, 50 mM Tris, pH 8.0) supplemented with protease inhibitor cocktail (1:500, Merck P8340). Lysates were centrifuged for 15 min at 10000 g, and protein concentration in the supernatant was determined by BCA assay (Merck B9643). Finally, 60 µg protein samples were separated using tricine SDS-PAGE[101] on 15% polyacrylamide gels and transferred to a PVDF membrane by semi-dry electroblotting (as described above). The incorporated radioactive methionine and cysteine signal was detected using Typhoon Imager (GE) and quantified using Image Lab 6 software (Bio-Rad). Afterwards, the membrane was probed with a specific antibody against actin (1:60000, Merck MAB1501) for normalisation of the radioactive signals of mitochondrial proteins.

## Whole Genome Sequencing

Genomic DNA was isolated from the frozen samples of hearts' left ventricles. Sequencing libraries were prepared using Illumina DNA PCR-Free Prep kit (Illumina, USA) and sequenced on NovaSeq X sequencer (Illumina, USA). The resulting fastq files were subjected to quality control and aligned to the Rattus norvegicus reference genome sequence(rn6) using bwa-mem2. After alignment, PCR duplicates was removed from the BAM files using Picard tools, base quality recalibration and variant identification was performed according to GATK best practices using the GATK package.

## Statistics and reproducibility

Data were analysed in GraphPad Prism 10 software (GraphPad Software). First, outliers were identified by ROUT method and then unpaired Student's t-test analysis or one sample t-test (two groups), one-way ANOVA (three and more groups) or two-way ANOVA (for comparisons of more than two parameters). Unless otherwise stated, the data shown are mean values ± S.D. of at least 3 independent experiments. A statistical difference of $p < 0.05$ was considered significant.

## Reporting summary

Further information on research design is available in the Nature Portfolio Reporting Summary linked to this article.

## Data availability

The published article and its supplementary information include all data generated and analysed in the study. Uncropped and unedited western blot images are provided in the Supplementary information as Supplementary Fig. 9. A complete source data file is provided as Supplementary Data. Label-free quantification mass spectrometry (LFQ-MS) data have been deposited at PRIDE and are publicly available (accession numbers: PXD045910 and PXD052976).

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

## Acknowledgements

This work was supported by the Czech Science Foundation GACR 19-10354S (P.P., V.K., G.P.-F., J.S., J.H., and A.P.), the National Institute for Research of Metabolic and Cardiovascular Diseases (Programme EXCELES, ID Project No. LX22NPO5104)—Funded by the European Union-Next Generation EU (K.T., T.M. and S.K.), and by the grant LUAUS23095 within the INTER-EXCELLENCE program of the Ministry of Education, Youth, and Sports of the Czech Republic (M.P.). O.G. was supported by the project P JAC CZ.02.01.01/00/22_008/0004575 RNA for therapy, Co-Funded by the European Union. The authors would like to acknowledge the Laboratory of Metabolomics at the Institute of Physiology of the Czech Academy of Sciences (T.C.) and the Proteomics Service Laboratory at the Institute of Physiology (supported by RVO, ID 67985823) and the Institute of Molecular Genetics (supported by RVO, ID 68378050) of the Czech Academy of Sciences (M.V.). The National Center for Medical Genomics (LM2023067) kindly provided WGS sequencing (S.K., V.S.). Graphical abstract and Figs. 1a and 5a were created with BioRender.com.

## Author contributions

Conceptualization, J.H., T.M., and A.P.; Methodology, P.P., M.P, T.M., and A.P.; Formal analysis, M.V., T.C., O.G., and V.S.; Investigation, P.P., K.C., V.K., G.P.-F., J.S., V.S., S.K., T.M., and A.P.; Resources, M.H., M.P., T.C., and M.V., Writing—Original draft, J.H., T.M., and A.P.; Writing—Review & Editing, P.P., K.C., V.K., K.T., M.V., T.C., O.G., M.P., J.H., T.M., and A.P.; Visualization, T.M. and A.P.; Funding acquisition, O.G., M.P., T.M., and A.P.

## Competing interests

The authors declare no competing interests.
