## [Peer Review File · Communications Biology]

Reviewers' comments:

Reviewer #1 (Remarks to the Author):

In this manuscript, Petr Pecina et al used a conplastic strains to investigate the contribution of mtDNA variants on metabolic perturbation. Although this remains the interest for me to see such work, many technical issues were raised and hinder me to give a positive feedback on this work in its current form.

Abstract

It is unclear to me how IR is caused by diacylglycerols accumulation. How does it work?

Introduction

The reference #2 is too old, and 1500 mitochondrial proteins seems not right. Also, I would suggest the authors to rewrite the introduction section while almost all the references cited in this section were 10-20 years ago, which is not the right way to draft introduction. Specifically, only a few words were used to describe the relationship between haplotype and metabolic syndrome, as well as haplotype and mitochondrial function.

Results

In Figure 1 data of body weigh should be presented in a range of time duration rather than end point. How did the energy intake was measured and it was not described in the method, I have to emphasize that energy intake/calorie intake is not equal with food intake. And the way to show OGTT is not right, we are looking for a time dependent dynamic change of glucose concentration when OGTT was started. Same problem was also found in figure 1F.

While there are too many issues need to be adjusted in almost all figure 1, I do not think it is necessary for me to continue review this manuscript, I am not sure if the authors can resubmit this manuscript, I would suggest the authors thoroughly revise this manuscript before submit back to comm Bio.

Reviewer #2 (Remarks to the Author):

Pecina et al. investigated the impact of mitochondrial DNA variation on metabolic physiology in rats by studying conplastic rats that carry different mtDNA from the SHR, BN, and F344 strains but share an identical nuclear background (SHR). The most interesting findings appear to be in the mtF344 strain, which exhibits (a) altered insulin signaling, (b) an increase in DGs in the liver, heart, and plasma, (c) an overall reduced oxygen consumption rate (FAO, OXPHOS, and ETC capacity), especially in HFD, and (d) reduced mitochondrial translation (thus OXPHOS subunit protein level) due to reduced mtSSU and possibly mtDNA sequence variance in mt-Rnr1 of the F344 strain.

Major:

(1) To confirm the specific mechanism in reduced mitochondrial translation and mt-Rnr1 in the mtF344 strain, more mitochondrial characterization might be needed. For instance, the mtDNA-encoded transcript level needs to be measured, and the protein and transcript level of other nuclear-encoded proteins that are involved in the mitochondrial central dogma.

(2) The author hypothesizes that the mtF344 phenotype is due to the mtDNA variants in the F344 strain, for which the same variants should be present in the original F344 strain. A comparison to the original F344 strain on a reduced OXPHOS complex and/or mtDNA-encoded protein translation rate (or level) might help to strengthen the conclusion.

Minors:

(1) Although very unlikely, the heteroplasmy in these mt strains needs to be shown. For instance, the content of mtDNA from mtF344 vs. mtDNA from SHR is indeed 100% vs. 0%.

(2) It remains unclear if the current strain method might still retain original nuclear genome from the other strain that could potentially explain the metabolic difference. The authors might want to comment on this possibility.

(3) As a unique model system, the author might want to expand the introduction and clearly explain these rat strains, the conplastic method, mtDNA variations among these strains, and previous findings on these rats.

(4) The reviewer felt that the conclusion that "glucose intolerance in the mtBN strain on HFD is associated with a selective decrease of complex IV" more based on the prior knowledge on mt-CO1 mutation, and not supported by oxygen consumption measurement shown in Fig. 5. Suggest reducing this part in the result.

Reviewer #3 (Remarks to the Author):

Reviewer Assessment

Manuscript#: COMMSBIO-23-4975 Reviewer: Antón Vila-Sanjurjo

Corresponding Author: Alena Pecinova Due Date: 27th Mar 24

Title: Haplotype variability in mitochondrial rRNA predisposes to metabolic syndrome

The manuscript entitled “Haplotype variability in mitochondrial rRNA predisposes to metabolic syndrome”, by Pecinova et al. studies the effect of mitochondrial sequence variation on the metabolic phenotype in conplastic rat strains. Their most important conclusion is that variation in the sequence of 12S mt-rRNA (2 variant residues, but possibly just one with phenotypic relevance) represents a risk factor in rats in a tissue- and diet-related fashion. The article is well written and provides ample experimental evidence, together with appropriate statistical support, to validate their claims.

In my opinion, this paper provides important support to the idea that primary sequence of mt-rRNA plays an significant role in mitochondrial functioning and disease. This novel, emergent idea is slowly gaining momentum in the field of mitochondrial research. Perhaps, for this reason, the authors have not completely realized how their research fits in the context of current scientific efforts in the field. It is, therefore imperative that the authors place their new findings in the proper scientific context of the state of the art (see Specific Issue 1, below). Besides enriching their manuscript, performing a cross-comparison to the relevant literature will aid in placing this emerging field of research in its proper light. Not doing so, would be a big disservice to the field.

The link between the two 12S mt-rRNA variants and the metabolic phenotypes is established via elegant biochemical and proteomics experiments that clearly show a defect in mitochondrial translation as the cause of the observed tissue- and diet-specific phenotypes. Unfortunately, the structural characterization of the two variants is, weak, at

best (see Specific Issue 4, below) and additional efforts are required in this regard, given the fact that high-resolution mito-ribosomal structures are available and relevant literature exists.

Specific Issues

1) The role of the mitochondrial translation machinery in disease is an emerging subfield within the vast field of mitochondrial research. Due to its novelty, it appears that many authors involved in this type research are ignorant of each other's contributions. In particular, the authors of this manuscript seem not to be aware of an important volume of relevant research performed in the last few years. The following reviews summarize the state of the field.

Vila-Sanjurjo A, Smith PM, Elson JL. Heterologous Inferential Analysis (HIA) and Other Emerging Concepts: In Understanding Mitochondrial Variation In Pathogenesis: There is no More Low-Hanging Fruit. *Methods Mol Biol.* 2021;2277:203-245. doi: 10.1007/978-1-0716-1270-5_14. PMID: 34080154.

Vila-Sanjurjo A, Mallo N, Atkins JF, Elson JL, Smith PM. Our current understanding of the toxicity of altered mito-ribosomal fidelity during mitochondrial protein synthesis: What can it tell us about human disease? *Front Physiol.* 2023 Jun 30;14:1082953. doi: 10.3389/fphys.2023.1082953. PMID: 37457031; PMCID: PMC10349377.

Additional related manuscripts that are relevant to specific conclusions raised by the authors are mentioned below. Aw et al. wrote a seminal article where they uncovered the effect of diet on the presentation of mtDNA variants in the context of *Drosophila* and hypothesized about the potential extension of such effects to mammals.

Aw WC, Towarnicki SG, Melvin RG, Youngson NA, Garvin MR, Hu Y, Nielsen S, Thomas T, Pickford R, Bustamante S, Vila-Sanjurjo A, Smyth GK, Ballard JWO. Genotype to phenotype: Diet-by-mitochondrial DNA haplotype interactions drive metabolic flexibility and organismal fitness. *PLoS Genet.* 2018 Nov 6;14(11):e1007735. doi: 10.1371/journal.pgen.1007735. PMID: 30399141; PMCID: PMC6219761.

The following paper describes the (very similar) biochemical effects caused by a disruptive mt-rRNA variant in a human cancer cell line. The reported conclusions must be compared to the submitted results:

Haumann S, Boix J, Knuever J, Bieling A, Vila Sanjurjo A, Elson JL, Blakely EL, Taylor RW, Riet N, Abken H, Kashkar H, Hornig-Do HT, Wiesner RJ. Mitochondrial DNA mutations induce mitochondrial biogenesis and increase the tumorigenic potential of Hodgkin and Reed-Sternberg cells. *Carcinogenesis*. 2020 Dec 31;41(12):1735-1745. doi: 10.1093/carcin/bgaa032. PMID: 32255484.

In the following article, the authors directly link a mt-rRNA variant to the effects of diet on fitness in *Drosophila*.

Dobson AJ, Voigt S, Kumpitsch L, Langer L, Voigt E, Ibrahim R, Dowling DK, Reinhardt K. Mitonuclear interactions shape both direct and parental effects of diet on fitness and involve a SNP in mitoribosomal 16s rRNA. *PLoS Biol*. 2023 Aug 21;21(8):e3002218. doi: 10.1371/journal.pbio.3002218. PMID: 37603597; PMCID: PMC10441796.

Studies on the effects of mitochondrial translation in disease in mammals were recently performed by the introduction of specific mito-ribosomal defects, namely mutated mitoribosomal proteins. Additionally, the effects of tissue and diet on the presentation of these mutations have been studied and must be compared to the submitted results. Such studies include:

Akbergenov R., Duscha S., Fritz A. K., Juskeviciene R., Oishi N., Schmitt K., et al. (2018). Mutant MRPS5 affects mitoribosomal accuracy and confers stress-related behavioral alterations. *EMBO Rep*. 19, e46193. 10.15252/embr.201846193

Ferreira N., Perks K. L., Rossetti G., Rudler D. L., Hughes L. A., Ermer J. A., et al. (2019). Stress signaling and cellular proliferation reverse the effects of mitochondrial mistranslation. *EMBO J*. 38, e102155. 10.15252/emboj.2019102155

Richman T. R., Ermer J. A., Siira S. J., Kuznetsova I., Brosnan C. A., Rossetti G., et al. (2021). Mitochondrial mistranslation modulated by metabolic stress causes cardiovascular disease and reduced lifespan. *Aging Cell* 20, e13408. 10.1111/accel.13408

Shcherbakov D., Juskeviciene R., Cortes Sanchon A., Brilkova M., Rehrauer H., Laczko E., et al. (2021). Mitochondrial mistranslation in brain provokes a metabolic response which mitigates the age-associated decline in mitochondrial gene expression. *Int. J. Mol. Sci*. 22, 2746. 10.3390/ijms22052746

Scherbakov D. S., Duscha S, , Juskeviciene R., 2nd, Restelli L., 3rd, Frank S., 4th, Laczko E., et al. (2020). Mitochondrial misreading in skeletal muscle accelerates metabolic aging and confers lipid accumulation and increased inflammation. *RNA* 27, 265–272.
10.1261/rna.077347.120

Additional potentially relevant papers:

Shcherbakov, D., Teo, Y., Boukari, H. et al. Ribosomal mistranslation leads to silencing of the unfolded protein response and increased mitochondrial biogenesis. *Commun Biol* 2, 381 (2019). <https://doi.org/10.1038/s42003-019-0626-9>

2) The position of the mt-rRNA variants is unclear. According to their Table S1 the variant positions are

mtF344 mtBN
648 NC A>C
935 A>G NC
942 C>T NC

but, according to Figure 7, the positions would be 868 and 875. Why this discrepancy?

In the same Figure 7C, they need to explain the colors used in the alignment and in the “identity” score, as well as the meaning of the asterisks and how gaps are indicated.

In Figure 7D, they must show the variant residues placed on the mito-ribosome closest to rat, which is likely to be mouse, according to their figure.

3) Variant nomenclature.

The authors should explicitly mention whether they are numbering their variant mt-rRNA positions according to gene or genome numbering. It seems that the positions given above follow gene numbering but this must be clearly stated. Since they will switch to genome numbering in the Discussion, namely when they describe the “C1518T polymorphism in human population mapping exactly to the same position as C942T polymorphism”, they probably should use both types of numbering to avoid confusion. Note that in the quoted sentence C1518T is human genomic numbering, whereas C942T is likely rat gene numbering, as discussed above. This lack of “official” nomenclature when naming mitochondrial variants only leads to confusion. For this reason, it is important to adhere to

what is emerging as the consensus, at least in humans, eg. “m.1518C>T”. Examples of how to use this nomenclature in the context of mt-rRNA can be found in

Vila-Sanjurjo A, Mallo N, Elson JL, Smith PM, Blakely EL, Taylor RW. Structural analysis of mitochondrial rRNA gene variants identified in patients with deafness. *Front Physiol.* 2023 Jun 8;14:1163496. doi: 10.3389/fphys.2023.1163496. PMID: 37362424; PMCID: PMC10285412.

It would be desirable that the authors adhered to this type of nomenclature to stay in concordance with relevant current literature and avoid confusion to the reader.

4) There are good reasons to believe that at least one of the variants identified by the authors can cause the observed effects. First, the biochemical and proteomics evidence presented by the authors supports the existence an important defect involving mitochondrial translation, more specifically the process of mt-SSU biogenesis, as reasonably discussed. However, how the actual base changes might lead to the observed defects has not been discussed in detail. The authors only mention the following:

“Interestingly, there is documented C1518T polymorphism in human population mapping exactly to the same position as C942T polymorphism in mtF344 rats, which occurs at frequency of 10^{-4} to 10^{-5} 66. However, its potential link to metabolic syndrome would have to be established.”

However, there is additional relevant information in the literature that can be used in order to provide, at least, an educated guess of the disruptive potential of the two variants present in mtF344 rats. For example, the effect of substitutions at the residue adjacent to m.1518C, namely at m.1517A (m.1517A>C variant), has been reported in

Vila-Sanjurjo A, Mallo N, Elson JL, Smith PM, Blakely EL, Taylor RW. Structural analysis of mitochondrial rRNA gene variants identified in patients with deafness. *Front Physiol.* 2023 Jun 8;14:1163496. doi: 10.3389/fphys.2023.1163496. PMID: 37362424; PMCID: PMC10285412.

and judged to be likely silent in the context of a mature mt-SSU. The same publication discusses the role of the human position m.1512A, which I believe it be the equivalent to their rat position m.935. In the tertiary structure of the human mito-ribosome, m.1512A is very close to position m.1537, where a C>U transition was determined to likely affect the functioning of bridge mB5 in h44. Hence, in addition to the potential assembly defect

during SSU biogenesis induced by the m.935A>G variant, which was effectively discussed by the authors, this substitution may potentially alter the functioning of bridge mB5 during translation.

5) In the Discussion section, the authors argue that the variable part of h44 where the two polymorphisms are located is part of the decoding center but this is not correct.

“The mt-Rnr1 polymorphisms identified in mtF344 are located in the variable part of the 3' minor domain helix h44 (Fig. 7c), a key decoding centre component.”

The decoding center is a loosely defined region surrounding the places where codon-anticodon interaction occurs at the A and P sites. Hence, the decoding center is far from the portion of h44 bearing the mutations.

Response to referees' comments regarding the manuscript COMMSBIO-23-4975: "Haplotype variability in mitochondrial rRNA predisposes to metabolic syndrome"

Referee expertise:

Referee #1: mitochondrial disease, mitochondrial genetics.

Referee #2: mitochondrial metabolism.

Referee #3: mitochondrial genetics, physiology.

Reviewers' comments:

Reviewer #1 (Remarks to the Author):

In this manuscript, Petr Pecina et al used a conplastic strains to investigate the contribution of mtDNA variants on metabolic perturbation. Although this remains the interest for me to see such work, many technical issues were raised and hinder me to give a positive feedback on this work in its current form.

Abstract

It is unclear to me how IR is caused by diacylglycerols accumulation. How does it work?

Introduction

The reference #2 is too old, and 1500 mitochondrial proteins seems not right. Also, I would suggest the authors to rewrite the introduction section while almost all the references cited in this section were 10-20 years ago, which is not the right way to draft introduction. Specifically, only a few words were used to describe the relationship between haplotype and metabolic syndrome, as well as haplotype and mitochondrial function.

Results

In Figure 1 data of body weigh should be presented in a range of time duration rather than end point. How did the energy intake was measured and it was not described in the method, I have to emphasize that energy intake/calorie intake is not equal with food intake. And the way to show OGTT is not right, we are looking for a time dependent dynamic change of glucose concentration when OGTT was started. Same problem was also found in figure 1F.

While there are too many issues need to be adjusted in almost all figure 1, I do not think it is necessary for me to continue review this manuscript, I am not sure if the authors can resubmit this manuscript, I would suggest the authors thoroughly revise this manuscript before submit back to comm Bio.

We are grateful for the suggestions to improve the manuscript, and we modified the manuscript according to most of the points raised by the reviewer:

The relation between diacylglycerol accumulation and IR is described in the Introduction, which we consider sufficient, as well as keeping the abstract text concise.

We updated the reference regarding the number of mitochondrial proteins referring to the most recent MitoCarta3.0 inventory (line 51):

"The nuclear genome encodes the remaining more than 1000 mitochondrial proteins, which are transported into the mitochondria²."

We included new supplementary figure S1 with body weight values during the time (Fig. S1a) as well as glucose plasma levels during the time (Fig. S1b).

Supplementary figure 1: Metabolic phenotype of the conplastic strains. (a) Body weights of mtSHR, mtBN and mtF344 animals during 15 weeks on chow (CHD) or high-fat (HFD) diet. The significant change between mtF344 and mtSHR on HFD diet is depicted as *a* and between mtBN and mtSHR as *b* (p -value < 0.05) **(b)** Time course of glycemia during oral glucose tolerance test (OGTT). Data represents means \pm S.D. from at least 9 animals.

Further, we changed the title of the y-axis in figure 1c (line 137) for more precise cumulative caloric intake and added the detailed method to **Materials and methods/Metabolic phenotype** (line 620). As HOMA-IR is calculated from fasting glucose and insulin values, the time-dependent dynamic change cannot be calculated.

Figure 1: Experimental design and metabolic phenotype of conplastic rats. (a) Schematic depiction of conplastic strains created by multiple backcrossing of males SHR rats with females of SHR, and females of progeny derived from (♀BN x ♂SHR)F1 or (♀F344 x ♂SHR)F1 hybrids. After weaning, experimental groups were transferred to a chow or high-fat diet for 15 weeks. At week 18, the OGTT test was performed, and at week 20, the tissues were collected for further analyses. Body weights (b) and cumulative caloric intake (c) after 15 weeks of dietary intervention. (d) Area under the curve calculated from oral glucose tolerance test. (e) Insulin levels 30 min after glucose gavage and (f) homeostatic model assessment (HOMA-IR) calculated from fasting glucose and insulin levels. Data represent means ± S.D. from at least 9 animals. Asterisks represent p-value: * <0.05; ** <0.01; *** <0.001 **** <0.0001.

Reviewer #2 (Remarks to the Author):

Pecina et al. investigated the impact of mitochondrial DNA variation on metabolic physiology in rats by studying conplastic rats that carry different mtDNA from the SHR, BN, and F344 strains but share an identical nuclear background (SHR). The most interesting findings appear to be in the mtF344 strain, which exhibits (a) altered insulin signaling, (b) an increase in DGs in the liver, heart, and plasma, (c) an overall reduced oxygen consumption rate (FAO, OXPHOS, and ETC capacity), especially in HFD, and (d) reduced mitochondrial translation (thus OXPHOS subunit protein level) due to reduced mtSSU and possibly mtDNA sequence variance in mt-Rnr1 of the F344 strain.

Major:

(1) To confirm the specific mechanism in reduced mitochondrial translation and mt-Rnr1 in the mtF344 strain, more mitochondrial characterization might be needed. For instance, the mtDNA-encoded transcript level needs to be measured, and the protein and transcript level of other nuclear-encoded proteins that are involved in the mitochondrial central dogma.

Thank you for the valuable suggestion. First, we measured the transcript levels of mtDNA-encoded genes *mt-Nd1*, *mt-Cyb*, *mt-Co1*, two genes encoding for the small mitochondrial ribosomal subunits whose protein levels were changed the most (*Mrps22*, *Mrps34*), and the gene *Mrpl37* that encodes for the subunit of the large mitochondrial ribosomal subunit. The transcript levels for individual genes were comparable or, in the case of *mt-Co1* and *Mrps22*, slightly but significantly increased in mtF344 animals on the high-fat diet. The results are included in the manuscript as supplementary figure S7a, b (see below). The new data indicate that changes in the content of mitochondrial ribosomes indeed occur in a posttranscriptional manner.

a)

b)

c)

Supplementary figure 7: Relative mRNA levels of mitochondrial encoded subunits and mitochondrial ribosomal proteins. Relative mRNA levels of (a) mitochondrial ribosomal proteins encoded by *Mrps22* and *Mrps34* (small mitochondrial ribosomal subunit) and *Mrpl37* (large mitochondrial ribosomal subunit) genes and (b) OXPHOS subunits encoded by *mt-Nd1*, *mt-Cyb* and *mt-Co1* genes in heart of mtF344 and mtSHR on CHD or HFD. (c) Relative mRNA levels of mitochondrial ribosomal proteins (as in a) in cultured skin fibroblasts derived from mtF344 and mtSHR. Data represents means \pm S.D. from at least 9 animals. Asterisks represent p-value: * < 0.05; ** < 0.01; *** < 0.001; **** < 0.0001.

Further, since we show decreased translation in skin fibroblasts derived from the mtSHR and mtF344 strains, we also measured the transcript levels of the same genes in these cells. Again, the mRNA levels were not changed significantly, and the results are included in figure 7 (line 389, see below) and supplementary figure S7c (see above):

Figure 7: Mitochondrial protein translation and the small mitoribosomal subunit. (a) Metabolic *in vivo* labelling with ³⁵S of mtDNA-encoded OXPHOS subunits after 3h with ³⁵S-methionine and ³⁵S-cysteine. Representative image shows autoradiographic detection of labelled proteins in 60 μg protein of whole cell lysates analysed by SDS-PAGE. mt-Nd1, 3, 4l and 6 – complex I; mt-Cyb – complex III; mt-Co1–3 – complex IV; mt-Atp6, 8 – complex V. Actin (Act) antibody was used as a loading control. (b) Relative mRNA level of OXPHOS subunits *mt-Nd1*, *mt-Co1* and *mt-Cyb*. (c) Representative Western blot analysis of the steady-state levels of subunits Ndufa9 (complex I); Uqcrc2 (complex III); Cox4 (complex IV), and Atp5f1a (complex V) normalized to Sdha (complex II). Actin IgM (Act) antibody was used as a loading control. Data represent means ± S.D. from 3 independent experiments. Asterisks represent p-value: * <0.05; ** <0.01; *** <0.001 **** <0.0001 (n = 3). (d) Multiple sequence alignment of h44 from selected species was generated by Clustal Omega⁴¹ and visualised in Geneious Prime (Biomatters Ltd.). Sites of single nucleotide polymorphisms in F344 mtDNA are highlighted in blue rectangles, reported positions of human pathogenic variants indicated by the red asterisks. (e) Structures of human (white, PDB 7po2)³⁵, mouse (pale blue, PDB 7pnw)³⁵ and porcine (grey, PDB 7nsi)⁴² small mitoribosomal subunits (intersubunit side) were superposed and visualised in ChimeraX⁴³. The helix h44 of human mtSSU rRNA is shown in yellow, and the variable region is red, the position of SNPs in 12S rRNA of F344 strain (m.935 and m.942) is depicted.

We modified the corresponding paragraph in the **Results/Lower oxidative capacity in the heart of mtF344 rats on HFD is caused by decreased content of OXPHOS complexes** (line 319):

...“To rule out the possibility, that the changed levels of mtSSU/LSU as well as OXPHOS proteins do not stem from the decreased transcript levels for individual proteins, we quantified transcripts of selected proteins from mtSSU (*Mrps22* and *Mrps34*), mtLSU (*Mrpl37*) and three mtDNA encoded OXPHOS subunits (*mt-Nd1*, *mt-Cyb* and *mt-Co1*). As depicted in Fig. S7, we did not observe any drop in the transcript content for mtDNA-encoded proteins. In the case of representative mitochondrial ribosomal subunits, we even observed a significant increase in the *Mrps22* transcript, possibly representing some sort of compensatory mechanism, since at the protein level, *Mrps22* was the most profoundly downregulated one (Fig. S7a). The data implicate that the OXPHOS proteins and mitochondrial content are generally suppressed due to the reduced amount of mtSSU that is not caused by lower expression of the mtSSU proteins, but rather the mitochondrial ribosomal subunits biogenesis is affected.”

Also, the part **Results/Downregulation of small mitochondrial ribosomal subunit proteins leads to the attenuation of mitochondrial proteosynthesis** has been modified (line 352):

...“Similarly to the heart tissue, the transcript levels of the mitochondrial ribosome genes were comparable between mtF344 and mtSHR animals. The transcript levels of three mtDNA encoded subunits were not changed (Fig. 7b) and the levels of all OXPHOS complexes with mtDNA-encoded subunits were slightly yet not significantly decreased in mtF344 fibroblasts (Fig. 7c). This suggests that the mitochondrial protein translation (not transcription) is slower but can be still sufficient in skin fibroblasts under non-stress conditions to achieve mitochondrial proteostasis.”

Accordingly, we modified **Materials and methods/Gene expression analysis** by adding the sample description and the gene expression assay numbers (line 642), and we clarified the corresponding parts in **Discussion**:

Line 491:

...“In the mtF344 strain, we demonstrated that mitochondrial mass, including OXPHOS proteins, is diminished in the heart of HFD-fed animals (Fig 6b, c and S5b), which was not a consequence of increased autophagy (Fig. S6a) or suppressed gene expression (Fig. S7b).”

Line 509:

...“Concomitantly, the mitochondrial protein synthesis was attenuated in skin fibroblasts derived from mtF344 animals compared to mtSHR (Fig. 7a), but the expression of the mitochondrial protein coding genes and mitochondrial ribosomal genes remained unchanged (Fig. 7b, S7c). Hence, the only genetic variability in components forming mtSSU remains rRNA constituent.”

(2) The author hypothesizes that the mtF344 phenotype is due to the mtDNA variants in the F344 strain, for which the same variants should be present in the original F344 strain. A comparison to the original F344 strain on a reduced OXPHOS complex and/or mtDNA-encoded protein translation rate (or level) might help to strengthen the conclusion.

We appreciate the suggestion. As a new additional experiment, we performed label-free quantification analysis of heart tissue of the progenitor F344 and SHR strains and observed the same pattern of downregulated mtSSU and upregulated mtLSU subunits when comparing F344 with SHR (see below). Since we only had access to samples from the progenitor F344 strain on the chow diet, we did not see any changes in OXPHOS subunits. Nevertheless, this is exactly in line with the phenotype observed for mtF344, and we believe it strengthens our conclusions in a way suggested by Reviewer #2.

We extended figure 6e (line 330) with the heatmap of the mtSSU/mtLSU proteins in the heart of F344 strain compared to mtSHR strain fed with a chow diet:

Figure 6: Analysis of protein levels in the liver and heart in mtF344 strain. LFQ-MS analysis of liver (a) and heart (b) of mtF344 compared to mtSHR control group. Data represent mean \pm SEM of all proteins, mitochondrial (Mito-proteins – MitoCarta3.0 annotated proteins) and OXPPOS (subunits of complex I–V). (c) Heat maps depicting the average log₂ fold-change of individual subunits of OXPPOS complexes between mtF344 and mtSHR groups in heart on HFD. (d) GO enrichment analysis using differentially expressed proteins in the heart of mtF344 compared to the mtSHR group on HFD. Numbers represent gene counts in particular GO term. (e) Heat maps depicting the average log₂ fold-change of individual proteins of mitochondrial small (mtSSU) or large (mtLSU) ribosome subunits between mtF344 compared to mtSHR (liver and heart tissue) and progenitor F344 and mtSHR

(heart tissue). The significances in (a) and (b) were calculated as one sample t-test compared to the mtSHR control group (n = 6). Asterisks represent p-value: * <0.05; ** <0.01; *** <0.001 **** <0.0001.

We modified the paragraph in **Results/Lower oxidative capacity in the heart of mtF344 rats on HFD is caused by decreased content of OXPHOS complexes** (line 314) as follows:

...“Strikingly, we observed the same pattern of downregulated mtSSU and upregulated mtLSU subunits also when comparing hearts from progenitor SHR and F344 strains (Fig. 6e). Since F344 progenitors and mtF344 only share mtDNA, it strongly implicates that polymorphisms in mtDNA and not nuclear background are responsible for the observed downregulation in the content of mtSSU subunits.”

The **Discussion** was also modified accordingly (line 505):

...“The key observation to dissect pathogenic mechanism was proteomic analysis that demonstrated reduced levels of the proteins constituting a small mitochondrial ribosomal subunit (mtSSU) in conplastic mtF344 animals in comparison to control mtSHR. Moreover, the comparison of progenitor F344 strain with control mtSHR showed analogous decrease of mtSSU proteins (Fig. 6e).”

The strain information was added to the **Materials and methods/Animals** (line 599).

...” Animal experiments were approved by the Institutional Animal Care and Use Committee and the Committee for Animal Protection of the Czech Academy of Sciences (Approval Number: 58/2021) in agreement with the Animal Protection Law of the Czech Republic, which is fully compatible with the guidelines of the European Community Council directives 2010/63/EU. All efforts were made to minimise animal suffering and to reduce the number of animals used. All animals were housed in controlled conditions (22 ± 2 °C, 12:12 hours of light-dark cycle) with access to water and respective diets. The strains used in the study include progenitor highly inbred spontaneously hypertensive rats (SHR) (SHR/OlaIpcv strain) and highly inbred F344 (F344/Crl) strain as well as conplastic strains derived by selective replacement of the mitochondrial genome of SHR strain with the mitochondrial genome of highly inbred strains of Fischer (F344) or Brown Norway (BN). The conplastic strains were generated by sequential backcrossing of F344 (53 backcrosses) or BN (59 backcrosses) females with SHR males.

The high number of backcrossing will ensure highly identical nuclear genomes in conplastic strains, since by 10th backcross generation nuclear genome of conplastic and recipient strain is approximately 99.8%^{28,30}. The ratio of heterozygous versus homozygous loci corresponds theoretically to 1.1×10^{-16} (mtF344) and 1.73×10^{-18} (mtBN). Whole Genome Sequencing analysis of heart samples from individual conplastic strains was performed to verify that the strains are homoplasmic for respective mtDNA variants and do not carry contaminating heteroplasmy from SHR progenitors (Fig. S8, Table S3). Thus, the three different strains harbour the SHR (mtSHR), F344 (mtF344) or Brown Norway (mtBN) mitochondrial genome on identical SHR nuclear genetic background^{27,29} (Fig. 1a). Also, the animals of progenitor F344 strain was used.”

Minors:

(1) Although very unlikely, the heteroplasmy in these mt strains needs to be shown. For instance, the content of mtDNA from mtF344 vs. mtDNA from SHR is indeed 100% vs. 0%.

We agree that it is very unlikely that the mtDNA from SHR strain is present in mtF344 animals, because mtDNA is exclusively maternally inherited. Therefore, we initially did not check for the homoplasmic nature of our conplastic strains. Nevertheless, to verify that no mtDNA heteroplasmy can be detected, we performed Whole Genome Sequencing of genomic DNA isolated from hearts' left ventricles, which is now included in the revised version of the manuscript. Of note, we received the same data from sequencing DNA isolated from spleens, which we decided not to include. In any case spleen data

confirm, that homoplasmic mtDNA status is present in both post-mitotic and proliferating tissues. We modified the **Materials and methods/Animals** (line 599) and added a new **Whole Genome Sequencing** paragraph describing mtDNA homoplasmy in all analyzed conplastic strains (line 788):

“Whole Genome Sequencing. Genomic DNA was isolated from the frozen samples of hearts’ left ventricles. Sequencing libraries were prepared using Illumina DNA PCR-Free Prep kit (Illumina, USA) and sequenced on NovaSeq X sequencer (Illumina, USA). The resulting fastq files were subjected to quality control and aligned to the Rattus norvegicus reference genome sequence(rn6) using bwa-mem2. After alignment, PCR duplicates was removed from the BAM files using Picard tools, base quality recalibration and variant identification was performed according to GATK best practices using the GATK package.”

The new Table S3 and Supplementary figure S8 (see below) was added to supplementary data:

Supplementary table 3: SNP variants unique for SHR strain - indicative of potential heteroplasmic contamination from the founder strain (N – nucleotide).

position No.		mtSHR			mtF344			mtBN		
SHR mtDNA	RN7 reference	N	coverage (reads)	identity (%)	N	coverage (reads)	identity (%)	N	coverage (reads)	identity (%)
1137	1137	C	239	100	A	384	100	A	570	100
1521	1521	A	478	99	G	357	100	G	627	100
1585	1585	C	519	100	T	367	100	T	668	100
1716	1716	C	463	100	T	340	100	T	556	100
4351	4352	G	499	100	A	390	100	A	544	100
4695	4696	A	443	100	G	387	100	G	590	100
5198	5200	G	412	100	A	288	100	A	508	100
5200	5202	G	407	100	A	289	100	A	502	100
5235	5237	A	390	99	T	258	100	T	486	100
5267	5269	C	386	100	G	243	100	G	499	100
6976	6978	A	495	100	G	408	100	G	616	100
8019	8021	A	501	100	G	387	100	G	591	100
10225	10227	C	360	100	T	295	100	T	418	100
11223	11225	A	259	100	G	216	100	G	302	100
11358	11360	A	235	100	G	197	100	G	318	100
14773	14775	A	453	100	G	386	100	G	599	100

Supplementary figure 8: Mapping of individual reads for mtDNA against rn6 reference genome (BN strain based).

(2) It remains unclear if the current strain method might still retain original nuclear genome from the other strain that could potentially explain the metabolic difference. The authors might want to comment on this possibility.

Usually, 99.8% identity of the nuclear genome is reached by the backcrossing of 10 generations, which is generally accepted by the scientific community as identical genomes. Since the backcross generation in the particular animals used for the study was 53 for mtF344 and 59 for mtBN, the identity is theoretically even higher and the ratio of heterozygous versus homozygous loci corresponds theoretically to 1.1×10^{-16} (mtF344) and 1.73×10^{-18} (mtBN). We added this comment to the **Materials and methods/Animals** (line 608):

...“The conplastic strains were generated by sequential backcrossing of F344 (53 backcrosses) or BN (59 backcrosses) females with SHR males.

The high number of backcrossing will ensure highly identical nuclear genomes in conplastic strains, since by 10th backcross generation nuclear genome of conplastic and recipient strain is approximately 99.8%^{28,30}. The ratio of heterozygous versus homozygous loci corresponds theoretically to 1.1×10^{-16} (mtF344) and 1.73×10^{-18} (mtBN).”

(3) As a unique model system, the author might want to expand the introduction and clearly explain these rat strains, the conplastic method, mtDNA variations among these strains, and previous findings on these rats.

We agree with the comment, and we extended the **Introduction** part clarifying the model generation and introducing the previous results on a high-fructose diet by changing the last two paragraphs (line 85) as follows:

...“**Therefore**, we asked whether the physiological variation in the mitochondrial DNA sequence may directly contribute to symptoms of metabolic syndrome. For this purpose, we used the unique model of conplastic rats carrying mtDNA from spontaneously hypertensive rat strain (SHR), Brown Norway

strain (BN) or Fischer strain (F344) on the identical nuclear background of SHR strain that is widely used as an animal model of the metabolic syndrome²⁷⁻³⁰. SHR, BN and F344 strains harbour different mtDNA variants, which represent different mtDNA haplogroups present across the palette of laboratory rat strains²⁸. The conplastic strains were derived by the multiple backcrossing of male SHR and female BN or F344 strain that allows to reach more than 99.8 % identity in nuclear genome and thus isolate the effect of naturally occurring variation in the mitochondrial genome²⁷⁻²⁹.

The previous studies demonstrated that the exposure of mtBN or mtF344 strain to high-fructose diet can promote systemic metabolic disturbances including glucose intolerance and/or elevated insulin levels during oral glucose tolerance test compared to control mtSHR strain^{27,29}. In particular, the Brown Norway mtDNA variant led to a selective decrease of cytochrome c oxidase at protein as well as enzyme activity levels²⁹. Furthermore, the analysis of metabolic phenotype of the left ventricles, muscle, and liver of mtF344 strain revealed reduced activity and content of several respiratory chain complexes. This associated with cardiac remodelling and changes in heart functional parameters compared to the mtSHR progenitor strain²⁷. Nevertheless, the direct link between the disturbances of mitochondrial function and metabolic phenotype has not been described, yet. In the current study, we found that both conplastic strains developed insulin resistance on a high-fat diet, which in the mtF344 animals can be explained by differences in mitochondrial 12S rRNA sequence, leading to reduced substrate oxidation and subsequent accumulation of bioactive lipids and insulin resistance.”

(4) The reviewer felt that the conclusion that "glucose intolerance in the mtBN strain on HFD is associated with a selective decrease of complex IV" more based on the prior knowledge on mt-CO1 mutation, and not supported by oxygen consumption measurement shown in Fig. 5. Suggest reducing this part in the result.

We agree with the comment. The new finding is that the insulin resistance development in the mtBN strain is diet-independent, however, the mechanism behind it is still unclear. As suggested, we removed the section “*Glucose intolerance in mtBN on HFD is associated with selective decrease of complex IV*” from **Results** (line 251 in the original manuscript version), as well as the original figure S5 and part of the **Discussion** (line 459 in the original manuscript). The end of the **Results/mtF344 rats on HFD have lower mitochondrial oxidative capacity in the heart** (line 262) has been modified:

...“Since we observed significant genotype-associated changes in respiratory parameters only in the mtF344 animals, in the subsequent experiments we only focussed on mtF344 and characterisation of molecular mechanisms underlying this phenotype.”

Reviewer #3 (Remarks to the Author):

Reviewer Assessment

Manuscript#: COMMSBIO-23-4975 Reviewer: Antón Vila-Sanjurjo

Corresponding Author: Alena Pecinova Due Date: 27th Mar 24

Title: Haplotype variability in mitochondrial rRNA predisposes to metabolic syndrome

The manuscript entitled “Haplotype variability in mitochondrial rRNA predisposes to metabolic syndrome”, by Pecinova et al. studies the effect of mitochondrial sequence variation on the metabolic phenotype in conplastic rat strains. Their most important conclusion is that variation in the sequence of 12S mt-rRNA (2 variant residues, but possibly just one with phenotypic relevance) represents a risk factor in rats in a tissue- and diet-related fashion. The article is well written and provides ample

experimental evidence, together with appropriate statistical support, to validate their claims.

In my opinion, this paper provides important support to the idea that primary sequence of mt-rRNA plays a significant role in mitochondrial functioning and disease. This novel, emergent idea is slowly gaining momentum in the field of mitochondrial research. Perhaps, for this reason, the authors have not completely realized how their research fits in the context of current scientific efforts in the field. It is, therefore imperative that the authors place their new findings in the proper scientific context of the state of the art (see Specific Issue 1, below). Besides enriching their manuscript, performing a cross-comparison to the relevant literature will aid in placing this emerging field of research in its proper light. Not doing so, would be a big disservice to the field.

The link between the two 12S mt-rRNA variants and the metabolic phenotypes is established via elegant biochemical and proteomics experiments that clearly show a defect in mitochondrial translation as the cause of the observed tissue- and diet-specific phenotypes. Unfortunately, the structural characterization of the two variants is, weak, at best (see Specific Issue 4, below) and additional efforts are required in this regard, given the fact that high-resolution mito-ribosomal structures are available and relevant literature exists.

We agree with the comments, we were not aware of some papers listed below. Referring to these publications definitely improved the text of the **Discussion**. In order to adhere to the journal guide for authors in terms of the manuscript length, we decided to discuss only selected topics.

Specific Issues

1) The role of the mitochondrial translation machinery in disease is an emerging subfield within the vast field of mitochondrial research. Due to its novelty, it appears that many authors involved in this type research are ignorant of each other's contributions. In particular, the authors of this manuscript seem not to be aware of an important volume of relevant research performed in the last few years. The following reviews summarize the state of the field.

Vila-Sanjurjo A, Smith PM, Elson JL. Heterologous Inferential Analysis (HIA) and Other Emerging Concepts: In Understanding Mitochondrial Variation In Pathogenesis: There is no More Low-Hanging Fruit. *Methods Mol Biol.* 2021;2277:203-245. doi: 10.1007/978-1-0716-1270-5_14. PMID: 34080154.

Vila-Sanjurjo A, Mallo N, Atkins JF, Elson JL, Smith PM. Our current understanding of the toxicity of altered mito-ribosomal fidelity during mitochondrial protein synthesis: What can it tell us about human disease? *Front Physiol.* 2023 Jun 30;14:1082953. doi: 10.3389/fphys.2023.1082953. PMID: 37457031; PMCID: PMC10349377.

Additional related manuscripts that are relevant to specific conclusions raised by the authors are mentioned below. Aw et al. wrote a seminal article where they uncovered the effect of diet on the presentation of mtDNA variants in the context of *Drosophila* and hypothesized about the potential extension of such effects to mammals.

Aw WC, Towarnicki SG, Melvin RG, Youngson NA, Garvin MR, Hu Y, Nielsen S, Thomas T, Pickford R, Bustamante S, Vila-Sanjurjo A, Smyth GK, Ballard JWO. Genotype to phenotype: Diet-by-mitochondrial DNA haplotype interactions drive metabolic flexibility and organismal fitness. *PLoS*

Genet. 2018 Nov 6;14(11):e1007735. doi: 10.1371/journal.pgen.1007735. PMID: 30399141; PMCID: PMC6219761.

The following paper describes the (very similar) biochemical effects caused by a disruptive mt-rRNA variant in a human cancer cell line. The reported conclusions must be compared to the submitted results:

Haumann S, Boix J, Knuever J, Bieling A, Vila Sanjurjo A, Elson JL, Blakely EL, Taylor RW, Riet N, Abken H, Kashkar H, Hornig-Do HT, Wiesner RJ. Mitochondrial DNA mutations induce mitochondrial biogenesis and increase the tumorigenic potential of Hodgkin and Reed-Sternberg cells. *Carcinogenesis*. 2020 Dec 31;41(12):1735-1745. doi: 10.1093/carcin/bgaa032. PMID: 32255484.

In the following article, the authors directly link a mt-rRNA variant to the effects of diet on fitness in *Drosophila*.

Dobson AJ, Voigt S, Kumpitsch L, Langer L, Voigt E, Ibrahim R, Dowling DK, Reinhardt K. Mitonuclear interactions shape both direct and parental effects of diet on fitness and involve a SNP in mitoribosomal 16s rRNA. *PLoS Biol*. 2023 Aug 21;21(8):e3002218. doi: 10.1371/journal.pbio.3002218. PMID: 37603597; PMCID: PMC10441796.

Studies on the effects of mitochondrial translation in disease in mammals were recently performed by the introduction of specific mito-ribosomal defects, namely mutated mito-ribosomal proteins. Additionally, the effects of tissue and diet on the presentation of these mutations have been studied and must be compared to the submitted results. Such studies include:

Akbergenov R., Duscha S., Fritz A. K., Juskeviciene R., Oishi N., Schmitt K., et al. (2018). Mutant MRPS5 affects mitoribosomal accuracy and confers stress-related behavioral alterations. *EMBO Rep*. 19, e46193. 10.15252/embr.201846193

Ferreira N., Perks K. L., Rossetti G., Rudler D. L., Hughes L. A., Ermer J. A., et al. (2019). Stress signaling and cellular proliferation reverse the effects of mitochondrial mistranslation. *EMBO J*. 38, e102155. 10.15252/emboj.2019102155

Richman T. R., Ermer J. A., Siira S. J., Kuznetsova I., Brosnan C. A., Rossetti G., et al. (2021). Mitochondrial mistranslation modulated by metabolic stress causes cardiovascular disease and reduced lifespan. *Aging Cell* 20, e13408. 10.1111/accel.13408

Shcherbakov D., Juskeviciene R., Cortes Sanchon A., Brilkova M., Rehrauer H., Laczko E., et al. (2021). Mitochondrial mistranslation in brain provokes a metabolic response which mitigates the age-associated decline in mitochondrial gene expression. *Int. J. Mol. Sci.* 22, 2746. 10.3390/ijms22052746

Scherbakov D. S., Duscha S., Juskeviciene R., 2nd, Restelli L., 3rd, Frank S., 4th, Laczko E., et al. (2020). Mitochondrial misreading in skeletal muscle accelerates metabolic aging and confers lipid accumulation and increased inflammation. *RNA* 27, 265–272. 10.1261/rna.077347.120

Additional potentially relevant papers:

Shcherbakov, D., Teo, Y., Boukari, H. et al. Ribosomal mistranslation leads to silencing of the unfolded protein response and increased mitochondrial biogenesis. *Commun Biol* 2, 381 (2019).

Based on the suggestions, we modified the **Discussion** thoroughly, mainly the second part has been rewritten to accommodate suggested recent literature on this topic (see below). We discussed the potential effect of the mtDNA variants on cellular metabolism and diabetic traits or their role in the disease. Also, the effect of mitochondrial translation aberrations on animal physiology, specifically pertaining to the tissue specificity, is discussed (line 505):

...“The key observation to dissect pathogenic mechanism was proteomic analysis that demonstrated reduced levels of the proteins constituting a small mitochondrial ribosomal subunit (mtSSU) in conplastic mtF344 animals in comparison to control mtSHR. Moreover, the comparison of progenitor F344 strain with control mtSHR showed analogous decrease of mtSSU proteins (Fig. 6e). Concomitantly, the mitochondrial protein synthesis was attenuated in skin fibroblasts derived from mtF344 animals compared to mtSHR (Fig. 7a), but the expression of the mitochondrial protein coding genes and mitochondrial ribosomal genes remained unchanged (Fig. 7b, S7c). Hence, the only genetic variability in components forming mtSSU remains rRNA constituent. Between mtSHR and mtF344 strains, the *mt-Rnr1* gene that encodes for 12S rRNA harbours two mismatches – m.935G>A (genome numbering, corresponds to gene number 868G>A) and m.942T>C(875T>C) (Table S2).

The two polymorphisms are located within the helix 44 of the 12S rRNA 3' minor domain (h44, Fig. 7d). In humans, h44 matures during the formation of decoding centre, the key functional region of the mtSSU, and its folding is regulated by GTPase NOA1 which hinders the interaction of h44 with the docking site. Dissociation of NOA1 allows h44 to adopt its mature conformation³⁷. The part of h44 at the base of its stem contributes to mRNA decoding during translation⁶⁴. However, the two polymorphisms are located into a distal variable region of h44 that forms two parallel stretches of single stranded RNA exposed towards the mtSSU surface on the intersubunit side (Fig. 7d, e)⁶⁵ and does not interact with any protein in mature mtSSU⁶⁶ or during translation⁶⁴. Hypothetically, the region could interact with an unknown regulator. In the conplastic mtF344 model the mitochondrial translation is slowed down due to the reduced protein (not transcript) level of mtSSU proteins implying possible role of h44 variable region in the biogenesis of mtSSU.

Both *MT-RNR1* and *MT-RNR2* represent the most constrained sequences within human mtDNA with highest percentage of invariable bases⁶⁷. Therefore, also pathogenic nucleotide substitutions in human *MT-RNR1* are relatively rare, and mostly associate with deafness. Only two (m.1555A>G and m.1494C>T) have been confirmed and several others have reported status in MITOMAP database (Fig. 7d, red asterisks) and they are the major contributors to aminoglycoside-induced and non-syndromic genetic hearing loss in patients with maternally inherited DEAFness, autism spectrum intellectual disability, possibly antiatherosclerotic⁶⁸. Transmitochondrial m.1555A>G cell lines also pointed to a role for mitochondrial protein synthesis⁶⁹, but in the absence of experimental models of mtDNA manipulation, the molecular mechanisms involved remained largely unknown⁷⁰⁻⁷². These variants localize to the same h44 helix of *MT-RNR1* as the polymorphisms identified in mtF344 rats, yet predominantly DEAFness polymorphisms involve evolutionary more conserved nucleotides (Fig. 7d, e). It is therefore feasible, that mutations at more conserved positions have more severe DEAFness associated phenotype, while those at positions with lower degree of conservation may contribute to metabolic syndrome.

Interestingly, there is documented m.1518C>T (871C>T) polymorphism in human population mapping exactly to the same position as m.942T>C polymorphism in mtF344 rats, which occurs at frequency of 10⁻⁴ to 10⁻⁵⁶⁸. However, its potential link to metabolic syndrome would have to be established. Further,

the analysis of mitochondrial rRNA variants in humans with deafness identified the mutation at position m.1517A>C (870A>C) that is neighbour to the m.1518C position. As the m.1517A is unpaired and not involved in any hydrogen bond, the variant was considered as silent mutation in the context of a mature mtSSU. The authors also performed structural analysis of the rRNA variants in the proximity of mito-ribosomal bridges, which revealed that m.1537C>T (890C>T) transition could affect the functioning of bridge mB5 in h44⁶⁵. Interestingly, the m.1512A (corresponds to m.935A in rats) residue is very close to m.1537C residue in tertiary structure, and the nucleotide transition may thus also influence the bridge functioning during translation. The potential link between *MT-RNR1* sequence variation and diabetic traits was suggested by the mitochondrial genome-wide association mapping of metabolomic phenotypes in humans. By this approach two prominent polymorphisms in *MT-RNR1* were identified (m.715G>A and m.856A>G), which associated with C2/C10:1 or SM (OH)C16:1/lysoPC a C28:1 ratio, respectively⁷³. All these metabolites have links to regulation of insulin secretion and may therefore be analogous to the phenotype we identified in mtF344 rats. Ultimately, it should also be mentioned, that within *MT-RNR1* sequence 16AA microprotein MOTS-c is encoded, which is exported from mitochondria and appears to affect the regulation of cellular metabolism and insulin action in age-related diseases, such as type 2 diabetes mellitus⁷⁴. However, this mechanism acts independently of mitochondrial protein synthesis.

In recent years, the research on the role of mtDNA variants in metabolic regulation and disease development accelerates^{36,75}. Aw et. al⁷⁶ studied the effect of diet in 4 strains of *D. melanogaster* with different mitochondrial genomes and standardized nuclear genome (mitotypes). They observed switch in the relative fitness in two mitotypes on high protein or high carbohydrate diet that was driven by the larval development and found that the changes are driven by naturally occurring point mutation in mtDNA encoded complex I subunit⁷⁶. The crosstalk between mitochondrial and nuclear genomes in response to dietary intervention in *D. melanogaster* with diverse mitonucleogenotypes was also studied by another group⁷⁷. They found that the SNP in the gene encoding for 16S rRNA (*mt:lrRNA*, m.13934C>T) is sufficient to elicit the fitness impacts of altering diets which suggest that the variation in mtDNA does not need to change protein sequence to interact with nuclear genome⁷⁷. In humans, the mutation in *MT-RNR2* encoding for 16S rRNA (m.1728G>A) was also associated with the increase of tumorigenic potential of Hodgkin lymphoma cells⁷⁸.

Further, the mouse model of mitoribosomal mistranslation created by homozygous knock-in of mutant *Mrps5* that increases the error frequency rate of mitochondrial protein synthesis was studied⁷⁹⁻⁸¹. The analysis of brain mitochondria revealed reduced basal and maximal respiration, suppressed levels of ATP and increased ROS generation⁷⁹. The mutation led to hearing loss and age and stress-related alterations in behavioural traits in affected animals⁷⁹. Moreover, the metabolic profiling of skeletal muscle demonstrated elevated levels of age-associated metabolites accompanied by increased glycolysis, pentose phosphate pathway and fatty acid synthesis indicating the alterations of specific bioenergetic processes in an age-dependent manner^{80,81}. Another mouse models with mutated mitochondrial ribosomal protein *Mrps12* leading to error-prone (K72I) or hyper-accurate (K71T) mitochondrial translation has been recently investigated^{82,83}. The hyper-accurate translation resulted in slightly lower levels of overall mitochondrial proteins that did not have defects and consequently mitochondrial stress response was not activated to rescue protein synthesis. Consequently, the OXPHOS function was compromised and resulted in cardiomyopathy. When the animals were exposed to high-fat diet, enhanced translational accuracy protected the liver from lipid accumulation, but the animals still developed hypertrophic cardiomyopathy and the lifespan was reduced. These results suggest that translational rate is more important than translational accuracy^{82,83}, which is in agreement with our findings where decreased rates of mitochondrial translation associate with decreased oxidative capacity, specifically affecting the heart.”

2) The position of the mt-rRNA variants is unclear. According to their Table S1 the variant positions are

mtF344 mtBN

648 NC A>C

935 A>G NC

942 C>T NC

but, according to Figure 7, the positions would be 868 and 875. Why this discrepancy?

In the same Figure 7C, they need to explain the colors used in the alignment and in the “identity” score, as well as the meaning of the asterisks and how gaps are indicated.

In Figure 7D, they must show the variant residues placed on the mito-ribosome closest to rat, which is likely to be mouse, according to their figure.

We are grateful for uncovering this discrepancy, it occurred due to different numbering (Table S2 – genome numbering, figure 7 – gene numbering). We modified Table S2, where the genome numbering in consensual nomenclature is indicated with respect to mtSHR sequence:

Supplementary table 2. Mitochondrial DNA variants in transfer and ribosomal RNA genes in mtF344 and mtBN strains compare to mtSHR. Nucleotide No.: position in mtSHR (genome numbering). NC – no change in sequence with respect to mtSHR sequence.

gene	Nucleotide No.	mtF344	mtBN
Phe	m.32A	del A	del A
	m.52C	ins A	ins A
Cys	m.5198G	G>A	G>A
	m.5200G	G>A	G>A
	m.5235A	A>T	A>T
Tyr	m.5267C	C>G	C>G
Asp	m.6976A	A>G	A>G
His	m.11540A	NC	A>G
Thr	m.15331A	A>G	NC
Pro	m.15398C	NC	C>T
mt-Rnr1	m.648C	NC	C>A
	m.935G	G>A	NC
	m.942T	T>C	NC
mt-Rnr2	m.1099C	NC	C>T
	m.1132C	del C	NC
	m.1137C	C>A	C>A
	m.1223G	G>A	NC
	m.1248T	T>C	NC
	m.1521A	A>G	A>G
	m.1585C	C>T	C>T
	m.1653T	ins AC	NC
	m.1693T	NC	T>C
	m.1716C	C>T	C>T
	m.1832A	A>G	NC
	m.1918A	NC	G>A
	m.2170C	C>T	NC
m.2647T	NC	del T	

Further, in figure 7d the nucleotide numbering was changed to genome numbering (line 389, see below):

Figure 7: Mitochondrial protein translation and the small mitoribosomal subunit. (a) Metabolic *in vivo* labelling with ³⁵S of mtDNA-encoded OXPHOS subunits after 3h with ³⁵S-methionine and ³⁵S-cysteine. Representative image shows autoradiographic detection of labelled proteins in 60 µg protein of whole cell lysates analysed by SDS-PAGE. mt-Nd1, 3, 4l and 6 – complex I; mt-Cyb – complex III; mt-Co1–3 – complex IV; mt-Atp6, 8 – complex V. Actin (Act) antibody was used as a loading control. (b) Relative mRNA level of OXPHOS subunits *mt-Nd1*, *mt-Co1* and *mt-Cyb*. (c) Representative Western blot analysis of the steady-state levels of subunits Ndufa9 (complex I); Uqcrc2 (complex III); Cox4 (complex IV), and Atp5f1a (complex V) normalized to Sdha (complex II). Actin IgM (Act) antibody was used as a loading control. Data represent means ± S.D. from 3 independent experiments. Asterisks represent p-value: * <0.05; ** <0.01; *** <0.001 **** <0.0001 (n = 3). (d) Multiple sequence alignment of h44 from selected species was generated by Clustal Omega⁴¹ and visualised in Geneious Prime (Biomatters Ltd.). Sites of single nucleotide polymorphisms in F344 mtDNA are highlighted in blue rectangles, reported positions of human pathogenic variants indicated by the red asterisks. (e) Structures of human (white, PDB 7po2)³⁵, mouse (pale blue, PDB 7pnw)³⁵ and porcine (grey, PDB 7nsi)⁴² small mitoribosomal subunits (intersubunit side) were superposed and visualised in ChimeraX⁴³. The helix h44 of human mtSSU rRNA is shown in yellow, and the variable region is red, the position of SNPs in 12S rRNA of F344 strain (m.935 and m.942) is depicted.

3) Variant nomenclature.

The authors should explicitly mention whether they are numbering their variant mt-rRNA positions according to gene or genome numbering. It seems that the positions given above follow gene numbering but this must be clearly stated. Since they will switch to genome numbering in the Discussion, namely when they describe the “C1518T polymorphism in human population mapping exactly to the same position as C942T polymorphism”, they probably should use both types of numbering to avoid confusion. Note that in the quoted sentence C1518T is human genomic numbering, whereas C942T is likely rat gene numbering, as discussed above. This lack of “official” nomenclature when naming mitochondrial variants only leads to confusion. For this reason, it is important to adhere to what is emerging as the consensus, at least in humans, eg. “m.1518C>T”. Examples of how to use this nomenclature in the context of mt-rRNA can be found in

Vila-Sanjurjo A, Mallo N, Elson JL, Smith PM, Blakely EL, Taylor RW. Structural analysis of mitochondrial rRNA gene variants identified in patients with deafness. *Front Physiol.* 2023 Jun 8;14:1163496. doi: 10.3389/fphys.2023.1163496. PMID: 37362424; PMCID: PMC10285412.

It would be desirable that the authors adhered to this type of nomenclature to stay in concordance with relevant current literature and avoid confusion to the reader.

We unified the nucleotide numbering in the manuscript according to the reviewer’s suggestion and we changed the numbering to genome numbering with gene numbering indicated in parenthesis and clearly stated the species (line 505, see the modified discussion above).

4) There are good reasons to believe that at least one of the variants identified by the authors can cause the observed effects. First, the biochemical and proteomics evidence presented by the authors supports the existence an important defect involving mitochondrial translation, more specifically the process of mt-SSU biogenesis, as reasonably discussed. However, how the actual base changes might lead to the observed defects has not been discussed in detail. The authors only mention the following:

“Interestingly, there is documented C1518T polymorphism in human population mapping exactly to the same position as C942T polymorphism in mtF344 rats, which occurs at frequency of 10⁻⁴ to 10⁻⁵ 66. However, its potential link to metabolic syndrome would have to be established.”

However, there is additional relevant information in the literature that can be used in order to provide, at least, an educated guess of the disruptive potential of the two variants present in mtF344 rats. For example, the effect of substitutions at the residue adjacent to m.1518C, namely at m.1517A (m.1517A>C variant), has been reported in

Vila-Sanjurjo A, Mallo N, Elson JL, Smith PM, Blakely EL, Taylor RW. Structural analysis of mitochondrial rRNA gene variants identified in patients with deafness. *Front Physiol.* 2023 Jun 8;14:1163496. doi: 10.3389/fphys.2023.1163496. PMID: 37362424; PMCID: PMC10285412.

and judged to be likely silent in the context of a mature mt-SSU. The same publication discusses the role of the human position m.1512A, which I believe it be the equivalent to their rat position m.935. In the tertiary structure of the human mito-ribosome, m.1512A is very close to position m.1537, where a C>U transition was determined to likely affect the functioning of bridge mB5 in h44. Hence, in addition to the potential assembly defect during SSU biogenesis induced by the m.935A>G variant,

which was effectively discussed by the authors, this substitution may potentially alter the functioning of bridge mB5 during translation.

Thank you for the suggestion, we included a detailed discussion on the possible impact of the residue change on mitochondrial translation (line 542).

...“However, its potential link to metabolic syndrome would have to be established. Further, the analysis of mitochondrial rRNA variants in humans with deafness identified the mutation at position m.1517A>C (870A>C) that is neighbour to the m.1518C position. As the m.1517A is unpaired and not involved in any hydrogen bond, the variant was considered as silent mutation in the context of a mature mtSSU. The authors also performed structural analysis of the rRNA variants in the proximity of mito-ribosomal bridges, which revealed that m.1537C>T (890C>T) transition could affect the functioning of bridge mB5 in h44⁶⁵. Interestingly, the m.1512A (corresponds to m.935A in rats) residue is very close to m.1537C residue in tertiary structure, and the nucleotide transition may thus also influence the bridge functioning during translation.”

5) In the Discussion section, the authors argue that the variable part of h44 where the two polymorphisms are located is part of the decoding center but this is not correct.

“The mt-Rnr1 polymorphisms identified in mtF344 are located in the variable part of the 3' minor domain helix h44 (Fig. 7c), a key decoding centre component.”

The decoding center is a loosely defined region surrounding the places where codon-anticodon interaction occurs at the A and P sites. Hence, the decoding center is far from the portion of h44 bearing the mutations.

The part of the discussion containing this inaccuracy was removed (line 516).

In addition to the manuscript changes according to the reviewers' suggestions, we made several further modifications of the text to further improve the manuscript:

The most significant modification is the addition of a new paragraph in the **Materials and methods** section – **SDS-PAGE/Western blotting** (line 752), which was missing in the original text.

“**SDS-PAGE and Western blotting.** Proteins separation under denaturing conditions was performed in heart homogenates or cultured fibroblasts using tricine-sodium dodecyl sulfate polyacrylamide gel electrophoresis (SDS-PAGE). The heart homogenates were mixed with the SLB buffer (sample lysis buffer; 2 % (v/v) 2-mercaptoethanol, 4 % (w/v) SDS, 50 mM Tris (pH 7.0), 10 % (v/v) glycerol, 0.02 % Coomassie Brilliant Blue R-250), and incubated at 65 °C for 10 min. The cells were washed with ice-cold PBS and harvested in ice-cold RIPA buffer (150 mM NaCl, 1 % Nonidet NP-40, 1 % sodium deoxycholate, 0.1 % SDS, 50 mM Tris, pH 8.0) supplemented with Protease Inhibitor Cocktail (1:500, Merck P8340) and benzonase[®] nuclease (1:1000, Merck 70664). Samples were centrifuged (10000 x g, 15 min), supernatant was mixed with the SLB buffer, and incubated at 65 °C for 10 min. 20 µg of tissue or cell samples were separated on 12 % polyacrylamide gels using the Mini-PROTEAN III apparatus (Bio-Rad, USA) and transferred to a polyvinylidene difluoride (PVDF) membrane (Immobilon FL 0.45 µm, Merck) by semi-dry electroblotting (0.8 mA/cm², 1 hour) using a Transblot SD apparatus (Bio-

Rad)¹⁰¹. Immunodetection was performed using the OXPPOS kit (Abcam, ab110412) or primary antibodies against LC3B (Abcam, ab48394), citrate synthase (Abcam, ab129095) and actin (Calbiochem, CP01-1EA), and fluorescent secondary antibodies AlexaFluor 680 (Life Technologies) and IRDye 800 (LI-COR). The signals were analysed and quantified by Image Lab software (Bio-Rad). Experiments were performed three times to assess the statistical significance of the results. “

We also added three more co-authors of the manuscript (Markéta Hlaváčková, Viktor Stránecký, Stanislav Kmoč), who were involved in performing additional experiments required by the reviewers (whole genome sequencing, proteomic analysis of the parental rat strain F344). All these changes including minor corrections are highlighted in yellow in the attached document (Pecina_CB_revised.pdf).

REVIEWERS' COMMENTS:

Reviewer #2 (Remarks to the Author):

The author provided substantial new experimental data and made comprehensive text edits, which successfully addressed all of my questions.

Reviewer #3 (Remarks to the Author):

I believe that all issues have been satisfactorily resolved by the authors.

[Editor's note: before accepting to review your manuscript, reviewer #3 disclosed their active involvement in the same area of research and asked whether they should recuse themselves from reviewing given that their comments, especially any potential requests to cite and discuss related literature, would be perceived as a conflict of interest. The editor appreciated the reviewer's candour and allowed them to review the paper.]